# Interaction between Teneurin-2 and microtubules via EB proteins provides a platform for GABAA receptor exocytosis

Sotaro Ichinose[1], Yoshihiro Susuki[1], Nobutake Hosoi[2], Ryosuke Kaneko[3,4], Mizuho Ebihara[1], Hirokazu Hirai[2], Hirohide Iwasaki[1]*

[1]Department of Anatomy, Gunma University Graduate School of Medicine, Gunma, Japan; [2]Department of Neurophysiology & Neural Repair, Gunma University Graduate School of Medicine, Gunma, Japan; [3]Bioresource Center, Gunma University Graduate School of Medicine, Gunma, Japan; [4]KOKORO-Biology Group, Laboratories for Integrated Biology, Graduate School of Frontier Biosciences, Osaka University, Osaka, Japan

**Abstract** Neurons form dense neural circuits by connecting to each other via synapses and exchange information through synaptic receptors to sustain brain activities. Excitatory postsynapses form and mature on spines composed predominantly of actin, while inhibitory synapses are formed directly on the shafts of dendrites where both actin and microtubules (MTs) are present. Thus, it is the accumulation of specific proteins that characterizes inhibitory synapses. In this study, we explored the mechanisms that enable efficient protein accumulation at inhibitory postsynapse. We found that some inhibitory synapses function to recruit the plus end of MTs. One of the synaptic organizers, Teneurin-2 (TEN2), tends to localize to such MT-rich synapses and recruits MTs to inhibitory postsynapses via interaction with MT plus-end tracking proteins EBs. This recruitment mechanism provides a platform for the exocytosis of GABA$_A$ receptors. These regulatory mechanisms could lead to a better understanding of the pathogenesis of disorders such as schizophrenia and autism, which are caused by excitatory/inhibitory (E/I) imbalances during synaptogenesis.

*For correspondence:
h-iwasaki@gunma-u.ac.jp

Competing interest: The authors declare that no competing interests exist.

## Editor's evaluation

Ichinose and coauthors investigate the mechanisms that contribute to building inhibitory synapses through differential protein release from microtubules. In their valuable study, they find that teneurin-2 plays a role in this process in cultured hippocampal neurons via EB1 using a variety of genetic and imaging methods. The methods, data, and analysis are solid, and the manuscript will be of interest to neuroscientists and cell biologists interested in intracellular trafficking and synapse maturation.

## Introduction

Neurons connect to each other via synapses to generate dense neural circuits and exchange information via synaptic receptors to perform various physiological functions. Axon guidance, synaptic specificity, and synaptogenesis are essential processes for establishing functional synapses (*Sanes and Zipursky, 2020*). Excitatory and inhibitory synapses play distinct roles in information transfer, and their coordination, known as the E/I balance, is crucial for proper brain function; disruptions of this balance can lead to disorders such as autism spectrum disorder and schizophrenia (*Maffei et al., 2017*). These two types of synapses differ significantly in their surrounding cytoskeletons. Excitatory

postsynapses mainly form and mature on characteristic structures called dendritic spines, which are composed predominantly of actin. Meanwhile, microtubules (MTs) enter almost only in an activity-dependent manner (*Gu et al., 2008*; *Hu et al., 2008*; *Jaworski et al., 2009*; *McVicker et al., 2016*).

In contrast, inhibitory synapses are formed directly on the shafts of dendrites, where both actin and MTs can be continuously present. This characteristic is believed to be shared both intra- and extra-synaptically. Therefore, the distinction between inhibitory postsynaptic and non-synaptic membranes is determined solely by the accumulation of inhibitory synapse-specific components, such as the GABA$_A$ receptor γ2 subunit and gephyrin. The major receptors at inhibitory synapses in the hippo-campus are GABA$_A$ and glycine receptors, which are transported by members of the kinesin super-family, such as KIF5 and KIF21, together with the scaffold protein gephyrin (*Labonté et al., 2014*; *Maas et al., 2009*; *Maas et al., 2006*; *Nakajima et al., 2012*; *Twelvetrees et al., 2010*). Receptors transported along MTs are exocytosed at different locations by a different mechanism from that of glutamate receptors and then move to the postsynaptic region by lateral diffusion (*Dahan et al., 2003*; *Gu et al., 2016*). After arriving at the postsynapse, the receptor binds to actin filaments via gephyrin, thereby preventing diffusion. When receptors are no longer necessary, they are endocy-tosed and transported away by dynein in an MT-dependent manner (*Fuhrmann et al., 2002*; *Kittler et al., 2000*). Thus, the dynamics of inhibitory postsynaptic components rely significantly on MTs. It is important to note that MTs themselves exhibit dynamic behavior, with their plus ends undergoing continuous polymerization and depolymerization (*Akhmanova and Steinmetz, 2015*). Consequently, whether the plus ends of MTs reach their intended destinations becomes a crucial issue for cargo delivery. The repeated accumulation and disappearance of inhibitory synapses at the same location suggests the existence of an 'intended destination' (*Villa et al., 2016*). However, the central mole-cule that defines the intended destination remains unknown. In addition, specific subunits of GABA$_A$ receptors are present in non-synaptic regions, contributing to tonic inhibition (*Glykys et al., 2008*). The mechanisms by which these specific components selectively target inhibitory synapses and avoid accumulating in non-synaptic regions are still unclear. Motivated by these intriguing dynamics, we investigated the underlying mechanism.

Teneurin-2 (TEN2) is a type II membrane protein whose C-terminal resides extracellularly for adhe-sion and is one of the synaptic organizers that induces synapse formation (*Li et al., 2018*). TEN2 has two alternative splicing forms: a splicing insertion-positive (SS+) form with seven amino acids inserted and a splicing insertion-negative (SS-) form with no insertion. SS+ is involved in inhibitory synapse formation, while SS- is involved in excitatory synapse formation. During excitatory synapse formation, the SS- of the presynapse shows specificity by binding to Latrophilin-2/3 in the postsyn-apse (*Sando et al., 2019*). On the other hand, SS+ has been shown to potentially bind to unknown binding partners during inhibitory synapse formation. However, the process of maturing inhibitory synapses is not well understood. Meanwhile, several studies have suggested interactions between the intracellular domain of TEN2 and cytoskeletal molecules. In mammals, the teneurin family has four paralogs (TEN1-4), while in *Drosophila*, there are two teneurin orthologs, ten-a and ten-m. Mutations in ten-a, present in the presynapse, result in abnormalities in MTs, while mutations in ten-m, present in the postsynapse, result in abnormalities in the spectrin skeleton (*Mosca et al., 2012*). In avian visual pathways, TEN2, which can interact with actin in the intracellular domain, is suggested to be expressed during periods that correspond with target recognition and synaptogenesis (*Rubin et al., 2002*). However, the specific mechanism by which interactions between these cytoskeletal molecules and synapse organizers contribute to synaptogenesis is not fully understood.

In this study, we demonstrate that the interaction between synaptic organizers TEN2 and MTs at inhibitory postsynapses provides a platform for the exocytosis of GABA$_A$ receptors. We also highlight the unique subsynaptic signaling systems present at inhibitory postsynapses, which facilitate efficient protein accumulation during synaptogenesis. Disruption of these accumulation systems can lead to an E/I imbalance and psychiatric disorders such as autism spectrum disorder and schizophrenia, making further research in this area highly promising.

## Results

### Inhibitory postsynapses are clustered into three types according to cytoskeletal molecules

We aimed to observe and classify the cytoskeletal states of inhibitory postsynapses. In this study, we defined postsynapses as those with immunostaining of gephyrin intensity above a certain threshold (*Figure 1—figure supplement 1A–D*). To visualize the diversity of the cytoskeleton at inhibitory postsynapses, we used anti- microtubule-associated protein 2 (MAP2) antibodies and phalloidin to visualize MTs and actin, respectively, in neurons cultured for 20 days in vitro (DIV20). We preliminarily found that there are three types of inhibitory postsynapses: MT-rich (cluster 1), actin-rich (cluster 3), and both low-level MT and actin (cluster 2) (*Figure 1A and B*).

For proteins to accumulate at inhibitory postsynapses, it is essential for cargo to be transported along MTs by kinesin, which is a plus-end directed motor protein (*Labonté et al., 2014*; *Nakajima et al., 2012*; *Twelvetrees et al., 2010*). Recently, there has been much interest in the regulatory mechanisms of kinesins, particularly at the plus end of MTs (*Guedes-Dias et al., 2019*; *Pawson et al., 2008*; *Qu et al., 2019*). Therefore, it is crucial to determine whether the MTs in MT-rich postsynapses have their plus-ends, minus-ends, or intermediate lattice parts. To address this, we attempted to determine the polarity of MTs at inhibitory postsynapses using live imaging of end-binding proteins (EBs), which can track the plus end of MTs. Furthermore, to prevent the redistribution of synaptic proteins due to overexpression, we visualized gephyrin by immunostaining after live imaging (*Figure 1—figure supplement 1E*). Interestingly, we found that pausing of EB3-EGFP comet was more likely to occur in the dendritic shaft region positive for gephyrin than in the region negative for gephyrin, both in anterograde and retrograde directions (*Figure 1C–H* and *Figure 1—figure supplement 1F*). These results suggest a mechanism by which the plus-ends of MTs are enriched at MT-rich inhibitory postsynapses.

### NLGN2 and TEN2 with EB binding motifs localize to MT-rich synapses

Live imaging of EB suggests the presence of MT recruiters at inhibitory synapses, which may regulate cargo transport by kinesins (*Guedes-Dias et al., 2019*; *Pawson et al., 2008*; *Qu et al., 2019*). Since the regulation of protein transport and accumulation is linked to the development of synapses, we considered the possibility that synaptic organizers could act as MT recruiters to promote synaptogenesis (*Figure 2—figure supplement 1A*). To narrow down the candidates of MT recruiters, we performed a motif search for possible binding to EB. This search was based on a previous proteomics study that explored the proteins present in the synaptic cleft (*Loh et al., 2016*). Proteins with two motifs proven to bind to EB1, SxφP, and LxxPTPφ in the intracellular domain were searched for in the results of the proteomic study (*Honnappa et al., 2009*; *Kumar et al., 2017*). After confirming the location of the motifs, we narrowed the list of seven proteins as candidate molecules (*Supplementary file 1*). Neuroligin-2 (NLGN2), immunoglobulin superfamily member 9B (IgSF9b), and TEN2 were tested among these candidates because of their functions as adhesion molecules and antibody availability (*Poulopoulos et al., 2009*; *Sando et al., 2019*; *Woo et al., 2013*).

Next, we performed four-color immunostaining of each candidate along with gephyrin, MAP2, and actin for cluster analysis at DIV 20 (*Figure 2B*). Using the uniform manifold approximate projection (UMAP) method, we reduced the three variables of synaptic area, MT intensity, and actin intensity. We found that the clustering results were consistent with the preliminary observations (*Figure 1B*), with postsynapse clustering according to whether they were enriched in MAP2 or actin (*Figure 2B–F*). The postsynaptic area was slightly higher in MT-rich synapse. When we calculated odds ratios for each cluster for NLGN2, IgSF9b, and TEN2, we found that inhibitory postsynapses that were positive for NLGN2 or TEN2 were more likely to be classified in clusters 1 and 3, respectively. However, there was no trend for IgSF9b with respect to classification (*Figure 2G and H*). When MAP2 intensity was compared using classical single-parameter comparison, only NLGN2 and TEN2 showed significant differences between postsynaptic positivity and negativity (*Figure 1—figure supplement 1B*). These results suggest that NLGN2 and TEN2 tend to be more abundant at MT-rich postsynapses and are likely MT recruiters.

To assess which is more suitable as an MT recruiter, NLGN2 or TEN2, we referred to previous electron microscopy (EM) studies. EM studies have shown that NLGN2 is observed near the center of synapses (*Takács et al., 2013*; *Uchigashima et al., 2016*). In contrast, MTs are not present near the center of synapses (*Gulley and Reese, 1981*; *Linsalata et al., 2014*). Therefore, TEN2 is more suitable

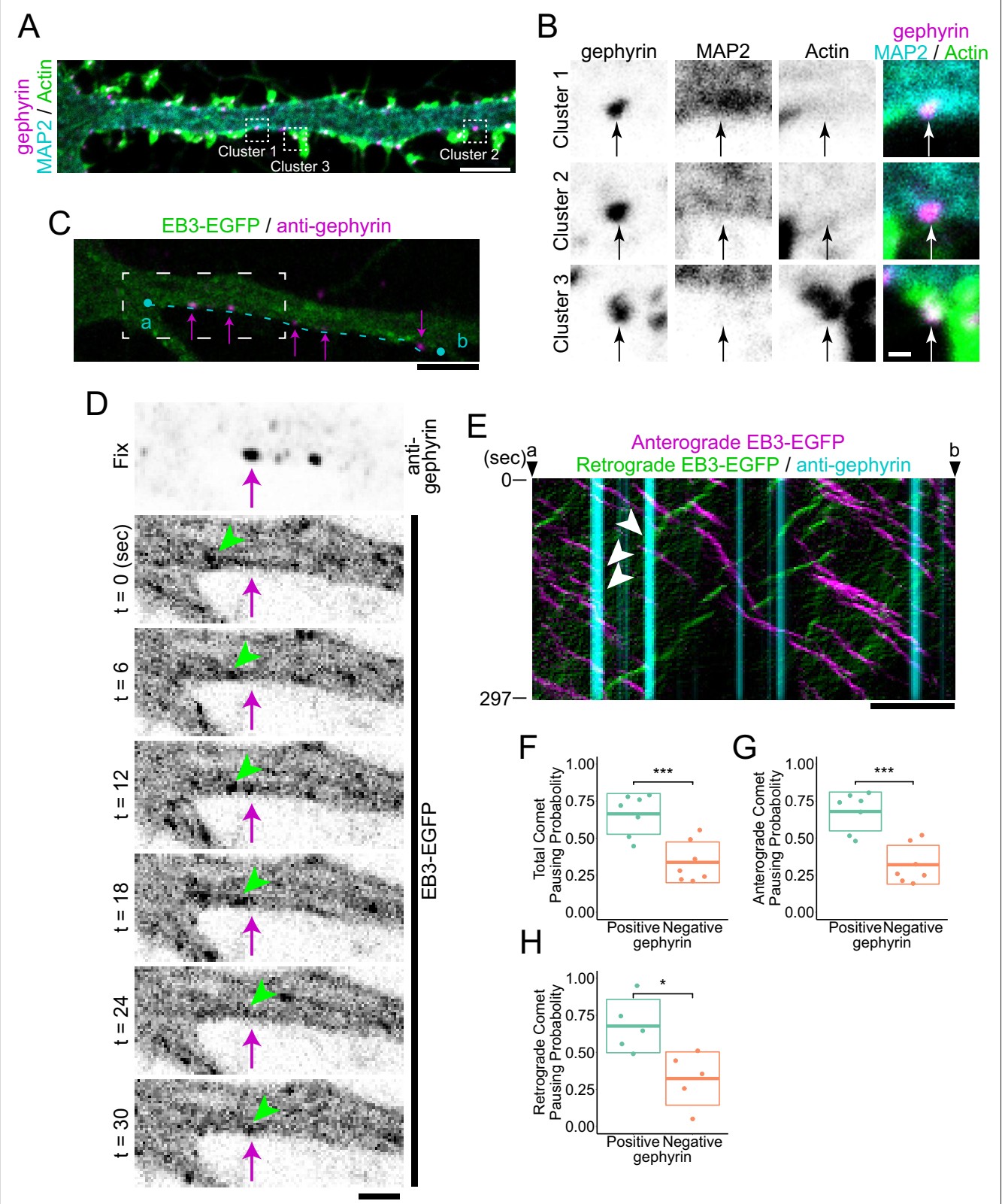

**Figure 1.** Cluster analysis of inhibitory postsynapses. (**A**) Image of immunostaining of gephyrin, MAP2, and actin. Cluster 1 is MT-rich synapses, cluster 2 is synapses with low levels of both MTs and actin, and Cluster 3 is actin-rich synapses. Typical synapses are boxed by dash lines with the cluster number attached to each, and an enlarged view is shown in (**B**). Scale bar, 5 μm. (**B**) Enlarged view of the synapses belonging to each cluster. Arrows indicate the position of postsynapses. Scale bar, 500 nm. (**C**) Overlaid images of live EB3-EGFP with the immunostained image of gephyrin. The timelapse image

*Figure 1 continued on next page*

*Figure 1 continued*

of the white dash line region is shown in (**D**). A kymograph of comets passing through an area 6.6 μm wide along the cyan dashed line between points a and b is shown in (**E**). Arrows indicate representative gephyrin positions. Scale bar, 5 μm. (**D**) Time-lapse imaging of EB3-EGFP and immunostained image of gephyrin. Arrows indicate the position of gephyrin. Arrowheads indicate tracking of a typical EB3 comet that dissipates at the position of gephyrin. Scale bar, 2 μm. (**E**) Kymograph of EB3-EGFP and gephyrin, with anterograde comets colored magenta and retrograde comets colored green. Arrowheads indicate typical EB3 comets that dissipate at the position of gephyrin. Scale bar, 5 μm. (**F–H**) Statistics of comet pausing probability. Total (**F**), anterograde (**G**), and retrograde (**H**) comets all had higher pausing probability at gephyrin-positive positions (p=8.0e-4 in F, p=2.5e-4 in G, p=0.014 in H by Welch's t-test). n=7 independent experiments. Two of the experiments were excluded from the statistics because a sufficient amount (>4) of retrograde comets were not observed (**H**). *p<0.05, ***p<0.001.

The online version of this article includes the following source data and figure supplement(s) for figure 1:

**Source data 1.** 4 Excel sheets containing the numerical data used to generate the *Figure 1F–H*.

**Figure supplement 1.** Cluster analysis of inhibitory postsynapses.

**Figure supplement 1—source data 1.** An Excel sheet containing the numerical data used to generate the *Figure 1—figure supplement 1B and C*.

than NLGN2. Thus, in this study, we focused on TEN2 and elucidated the mechanism by which MTs are recruited to inhibitory postsynapses.

## TEN2 is expressed on the surface of the inhibitory postsynapse during early synaptogenesis

TEN2 is a transmembrane protein present at synapses and has been reported to have a close relationship with cytoskeletal formation. However, its precise localization and function remains controversial (*Mosca et al., 2012*; *Sando et al., 2019*; *Silva et al., 2011*). To avoid misidentification due to antibody differences, we generated knock-in mice using the CRISPR/Cas9 system with a 3×HA tag inserted just before the STOP codon in exon 29 of TEN2, in addition to generating specific antibodies against the intracellular domain (anti-ICD) (*Figure 3—figure supplement 1A–D*). We co-stained primary hippocampal neuronal cultures prepared from this mouse with anti-ICD and anti-HA antibodies (*Figure 3—figure supplement 1E*). The Manders' overlap coefficient indicated that co-localization by these two antibodies was moderate, but both antibodies colocalized with gephyrin (*Figure 3—figure supplement 1F and G*). Moreover, there was no significant difference in the number of synapses identified as positive for TEN2 by each antibody, indicating that either antibody can be used to evaluate TEN2-positive inhibitory synapses. We then evaluated the expression of TEN2 during neuronal development using knock-in neurons. The results showed that detectable levels of TEN2 were not expressed by DIV4, when axon-dendrite polarity is formed, or by DIV7, when dendritic growth is activated (*Figure 3A* ). By DIV12, when initial synapse formation occurs, TEN2 began to be expressed, and most of the TEN2 at this time was surface-expressed (*Figure 3B*). Although some TEN2 remained intracellularly in the cell body and proximal dendrites, almost all TEN2 appeared on the extracellular surface in the middle and distal dendrites (*Figure 3B-D*). This trend was also observed in DIV15. Furthermore, surface-expressed TEN2 co-localized with gephyrin (*Figure 3—figure supplement 1I*). TEN2 was also found to localize within the spine visualized by phalloidin staining and partially co-localize with PSD-95, an excitatory synaptic marker, confirming previous studies that TEN2 is a molecule that functions at excitatory synapses (*Figure 3—figure supplement 1J–O*). These results suggest that TEN2 begins to be expressed early in synaptogenesis and that most of it is located at the surface of the plasma membrane, including the synaptic site.

Conventional microscopy has limited resolution, making it impossible to determine whether TEN2 is located in the presynaptic or postsynaptic membrane of inhibitory synapses. To overcome this resolution problem, we utilized stochastic optical reconstruction microscopy (STORM), a super-resolution microscopy (SRM) technique, to observe precise localization. First, to investigate whether TEN2 is present in the inhibitory presynapse or postsynapse, we co-stained cells with anti-ICD antibody and the presynaptic marker Bassoon. In STORM images, there was little overlap between TEN2 and Bassoon (*Figure 3E*). However, this result alone cannot rule out the presence of TEN2 in the presynaptic side because the size of the antibody is close to the synaptic cleft (~20 nm), and it does not distinguish between excitatory and inhibitory synapses. Therefore, we next performed double staining with vGAT, a marker of inhibitory presynapses. vGAT is a membrane protein present in GABA-containing synaptic vesicles, while Bassoon functions as a scaffolding protein in the active zone. By observing vGlut, a

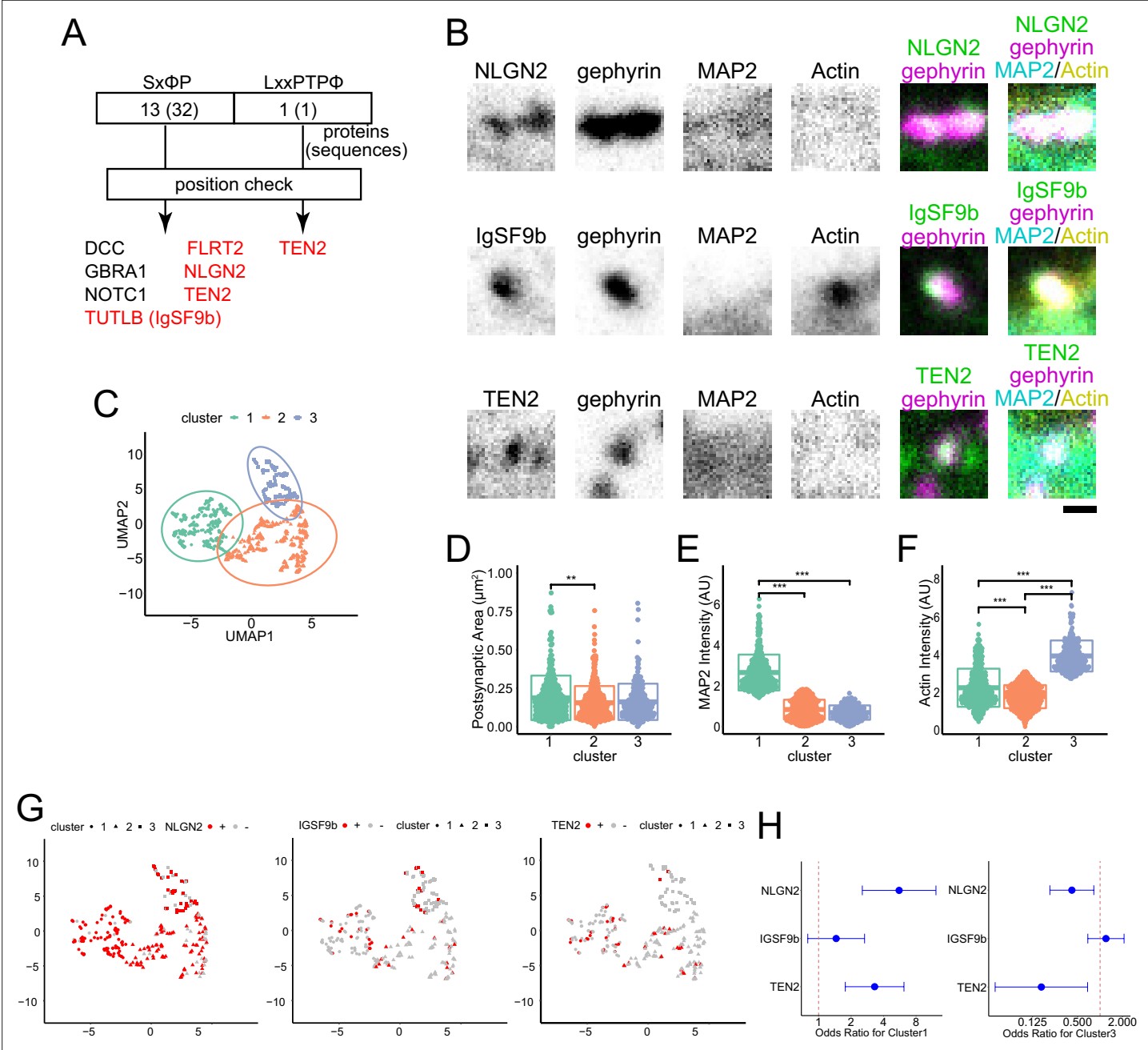

**Figure 2.** NLGN2 and TEN2 with EB binding motifs localize to MT-rich synapses. (**A**) Motif search results: SxφP motifs were found in 32 locations in 13 proteins; LxxPTPφ motifs were found in 1 protein. After checking whether these sequences are intracellular or extracellular, the number of candidate proteins was narrowed down to 7. Of these, those belonging to the adhesion molecule are shown in red. (**B**) Representative immunostained images of each synaptic organizer and gephyrin, MAP2, and actinin in DIV20 hippocampal cultured neurons. Scale bar, 500 nm. (**C**) Plots showing the results of cluster analysis. Three-dimensional parameters of synaptic area, MAP2 intensity, and actin intensity evaluated inhibitory postsynapses. After being reduced to two dimensions by UMAP, cluster analysis was performed with the number of clusters pre-specified as 3. The number of synapses belonging to each cluster was 315, 413, and 212 observed by three independent experiments. (**D–F**) Comparison between clusters for each parameter. (**D**) Synaptic area: One-way ANOVA showed a significant difference (p=0.0019), and Tukey multiple comparisons showed a significant difference between clusters 1 and 2 (p=0.0016). (**E**) MAP2 intensity: One-way ANOVA showed a significant difference (p<2e-16), and Tukey multiple comparisons showed significant differences between clusters 1 and 2 (p<1e-07) and between clusters 1 and 3 (p<1e-07). (**F**) Actin intensity: One-way ANOVA showed a significant difference (p<2e-16), Tukey multiple comparisons showed significant differences between clusters 1 and 2 (p<1e-07), between clusters 2 and 3 (p<1e-07) and between clusters 1 and 3 (p<1e-07). The sample size is the same as (**C**). **p<0.01, ***p<0.001. (**G**) Cluster analysis and the relationship between the positivity and negativity of each adhesion molecule. The calculation results by UMAP are the same as in (**C**). The number of NLGN2 positive and negative synapses are 228 and 65. The number of IgSF9b positive and negative synapses are 53 and 283. The number of TEN2 positive and negative

*Figure 2 continued on next page*

*Figure 2 continued*

synapses are 49 and 262. TEN2 positive had very little classification to Cluster 3, only 2 synapses. (**H**) The odds ratio and 95% confidence interval for each adhesion molecule for clusters 1 and 3. For cluster 1: NLGN2, 5.57 (2.54–12.2); IgSF9b, 1.45 (0.80–2.66); TEN2, 3.30 (1.77–6.17). For cluster 3: NLGN2, 0.42 (0.21–0.82); IgSF9b, 1.20 (0.69–2.09); TEN2, 0.16 (0.04–0.68).

The online version of this article includes the following source data and figure supplement(s) for figure 2:

**Source data 1.** 4 Excel sheets containing the numerical data used to generate the *Figure 2C–H*.

**Figure supplement 1.** NLGN2 and TEN2 with EB binding motifs localize to MT-rich synapses.

**Figure supplement 1—source data 1.** An Excel sheet containing the numerical data used to generate the *Figure 2—figure supplement 1B*.

membrane protein present in glutamate-containing synaptic vesicles, and Bassoon using dSTORM, the respective localizations near the presynaptic center and just below the membrane could be distinguished in the previous report (*Andreska et al., 2014*). This suggests that vGAT as well as vGlut can be used to determine presynaptic location more clearly than Bassoon. Thus, vGAT staining can solve both the problem of antibody size and the problem of distinguishing between excitatory and inhibitory synapses. Even in the double-stained STORM images, little overlap between TEN2 and vGAT was observed. In contrast, overlap between TEN2 and gephyrin was observed, indicating the presence of TEN2 at inhibitory postsynapses (*Figure 3E*). To confirm this result, we performed a proximity ligation assay (PLA) (*Söderberg et al., 2006*). In this assay, two antibodies were immunostained, and when they were in proximity (~20 nm), the oligonucleotides fused to the antibodies were ligated to generate circular DNA. Proximity was detected by incorporating a dye into dNTP. The PLA results showed that the proximity of TEN2 and gephyrin was significantly greater than that of normal IgG and gephyrin used as a negative control (*Figure 3H and I*). Furthermore, to confirm the abundance of TEN2 in dendrites, we performed mixed cultures of knock-in and wildtype neurons and stained them with an anti-HA antibody. The signal intensity of HA was reduced in wildtype dendrites that do not express TEN2 in the cell body, confirming that TEN2 is a protein that is abundant in dendrites (*Figure 3—figure supplement 1P and Q*). The fact that the signal did not reach zero in wildtype dendrites suggests that TEN2 is present in small amounts in axons surrounding dendrites. These results support the dSTORM data and suggest that our dSTORM images are not affected by signal misalignment between channels due to drift or chromatic aberration. Interestingly, the puncta of TEN2 and gephyrin were not always perfectly colocalized. Therefore, we measured the distance between the centroids of the fluorescence intensity of each punctum and found that they were 85 nm apart (*Figure 3F and G*). Considering the width of the inhibitory postsynapse (approximately 500 nm), this distance is not far from the perisynaptic area. Alternatively, this may suggest that TEN2 is specific to a particular nanodomain in the synapse (*Yang et al., 2021*). These results suggest that TEN2 is more abundant at inhibitory postsynapses and is located primarily away from the center of the synapse in nano-scale.

## TEN2 provides a platform for the exocytosis of GABA$_A$ receptors at inhibitory postsynapses to mature synapses

TEN2 overexpression in non-neuronal cells induces formation of both excitatory and inhibitory synapses in attached neurons (*Sando et al., 2019*). To determine whether TEN2 also induces synapse formation in neuron-neuron interactions, we knocked down TEN2 in primary hippocampal cultures using RNA interference (RNAi). Knockdown was achieved using a vector-based shRNA that contained an shRNA sequence and a fluorescence protein sequence, allowing for the expression of a fluorescence protein as a volume marker. The half-life of TEN2 in rat hippocampal primary culture neurons from DIV11 was reported to be 1.42 days (*Heo et al., 2018*). Based on these data, we transfected DIV11 cells with the TEN2 knockdown vector and performed fixation and immunostaining 3 days after transfection. We quantified the amount of TEN2 expression in the cell body using immunostaining with anti-ICD antibody and confirmed knockdown (*Figure 4A and B*). Under these conditions, we found that the number of gephyrin puncta was significantly reduced in knockdown neurons, correlating with the remaining amount of TEN2, suggesting that postsynaptic TEN2 is involved in the formation of inhibitory postsynapses (*Figure 4C and D*).

We next confirmed the function of TEN2 on inhibitory synapses by evaluating its effect on GABA$_A$ receptors. There are 19 subunits of GABA$_A$ receptors, and those localized to inhibitory synapses form a heteropentamer consisting of two α1–3 subunits, two β1–3 subunits, and one γ2 subunit, arranged

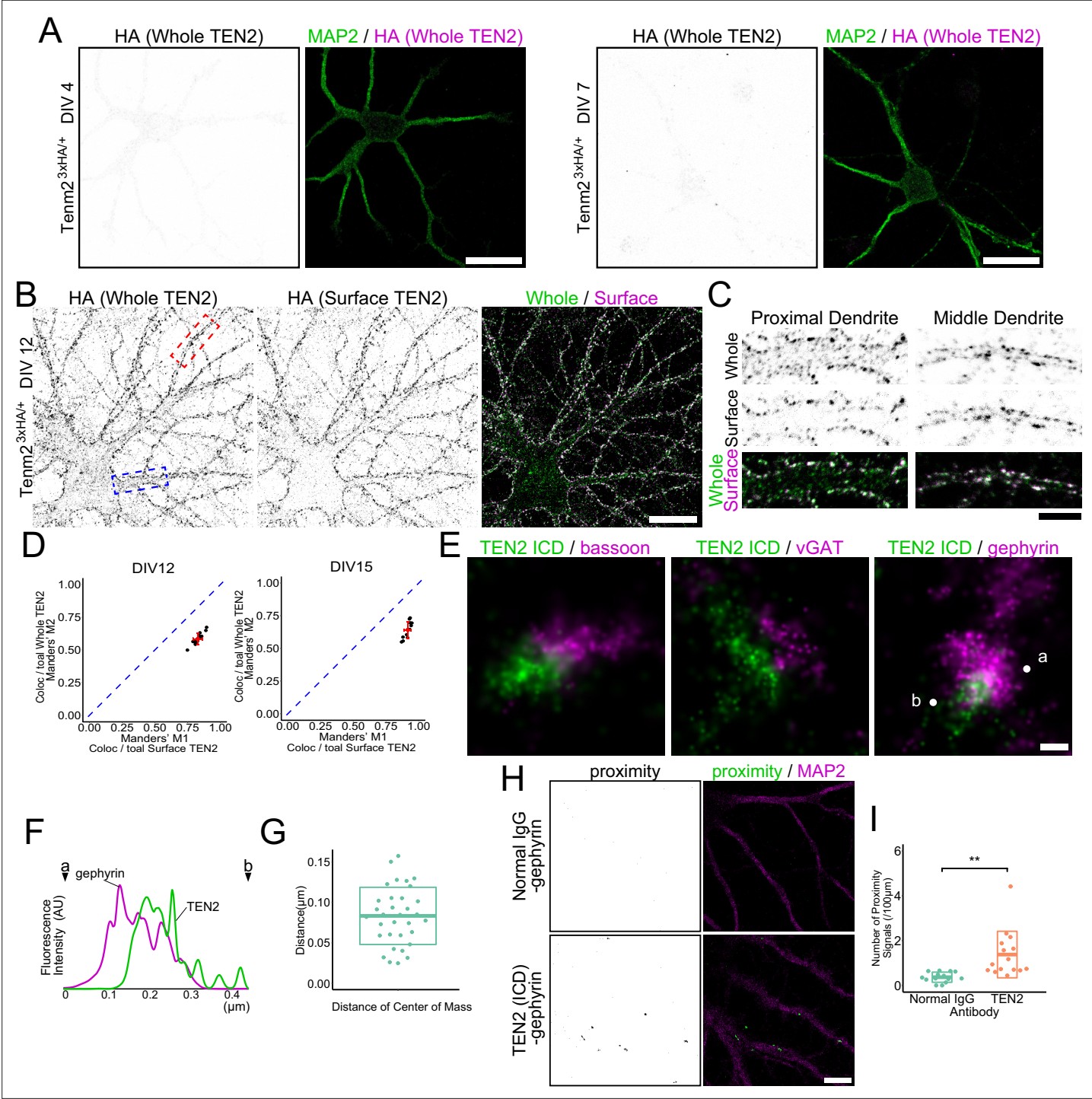

**Figure 3.** TEN2 is expressed on the surface of the inhibitory postsynapse during early synaptogenesis. (**A**) Low expression of TEN2 in early neural development. TEN2 is not expressed in detectable amounts at DIV4 and DIV7. Scale bars, 20 μm. (**B**) Surface expressed TEN2. TEN2 is expressed, and most TEN2 are surface expressed at DIV12. The area boxed by the dashed line is shown in (**C**). The blue box indicates the proximal dendrite, and the red box indicates the middle dendrite. Scale bar, 20 μm. (**C**) Immunostaining of whole TEN2 and surface TEN2 in proximal and middle dendrites. TEN2 is observed intracellularly in the proximal dendrite, while most TEN2 was surface expressed in the middle dendrite. Scale bar, 5 μm. (**D**) Statistical analysis of whole TEN2 and surface TEN2 measurements. Plots were generated to visualize the values, and red crossbars represent the mean ± SD. The blue dashed line represents M1=M2, which indicates equality according to Manders' overlap coefficient. For DIV12, the Mander's coefficient values were M1, 0.83±0.037; M2, 0.58±0.044. n=15 neurons. For DIV15, the Mander's coefficient values were M1, 0.92±0.026; M2, 0.64±0.065. n=12 neurons. (**E**) dSTORM images. Two-color staining of each presynaptic and postsynaptic molecule suggests that TEN2 is more abundant in the postsynapses.

*Figure 3 continued on next page*

*Figure 3 continued*

Scale bar, 100 nm. (**F**) Line graph showing the signal intensity of TEN2 and gephyrin. The horizontal axis shows the length, and the vertical axis shows the fluorescence intensity. Points indicated by letters and arrowheads represent the positions of 'a' and 'b' in (**E**). (**G**) Distance between the centers of mass of TEN2 and gephyrin when observed in dSTORM. The mean ± SD was 83.3±35.3. n=33 synapses. (**H**) Images showing the results of the proximity ligation assay. When the proximity ligation assay was performed using antibodies against TEN2 and gephyrin, a signal indicating the proximity of less than 20 nm could be detected. On the other hand, no signal was obtained in the negative control. Scale bar, 10 µm. (**I**) The number of proximity signals per 100 µm. mean ± SD was 0.37±0.23 and 1.38±1.04, respectively. Welch's t-test showed a significant difference between negative control and TEN2 in proximity to gephyrin (p=0.0021). n=14 and 15 from three independent experiments. **p<0.01.

The online version of this article includes the following source data and figure supplement(s) for figure 3:

**Source data 1.** 4 Excel sheets containing the numerical data used to generate the *Figure 3D, F, G and I*.

**Figure supplement 1.** TEN2 is expressed on the surface of the inhibitory postsynapse during early synaptogenesis.

**Figure supplement 1—source data 1.** Unprocessed full-size gel photograph showing genotyping of knock-in mice and photograph showing the region used in *Figure 3—figure supplement 1C* with dashed lines.

**Figure supplement 1—source data 2.** 3 Excel sheets containing the numerical data used to generate the *Figure 3—figure supplement 1G, H and Q*.

counterclockwise from the extracellular side as γ2-β-α-β-α. Quantification of α1, α5, and γ2 subunits was performed. However, α5 occasionally formed elongated clusters and could not be quantified as the number of puncta, so it was quantified as the total amount of fluorescence. As a result, only the expression of γ2 was significantly reduced by TEN2 knockdown (*Figure 4E and F*). As γ2 is present in all synaptic GABA$_A$ receptors, the significant decrease in gephyrin is thought to be related to the reduction in γ2.

To confirm that TEN2 promotes the accumulation of GABA$_A$ receptors, we performed a fluorescence recovery after photobleaching (FRAP) assay. We visualized the surface expression of GABA$_A$ receptors by expressing a very small amount of receptors in which pHluorin was inserted into the extracellular domain of the γ2 subunit (*Jacob et al., 2005*). By performing photobleaching over a wide area and observing fluorescence recovery, we monitored the exocytosis of GABA$_A$ receptors after photobleaching (*Figure 4G*). The puncta-shaped pHluorin recovered, supporting previous studies that showed that the accumulation of GABA$_A$ receptors occurs through exocytosis after transport by kinesin, rather than lateral diffusion from the cell body (*Nakajima et al., 2012*; *Twelvetrees et al., 2010*). We further determined whether each punctum was TEN2-positive or -negative by fixing the neurons after the FRAP assay and visualizing TEN2 localization using an anti-HA antibody. The results showed that pHluorin puncta positive for TEN2 had significantly higher fluorescence recovery than TEN2-negative puncta (*Figure 4H–J*). These results suggest that TEN2 is involved in the postsynaptic formation of inhibitory synapses by providing a platform for the exocytosis of GABA$_A$ receptors.

## Postsynaptic TEN2 knockdown affects inhibitory synapses functionally

To investigate the functional role of TEN2 in hippocampal neurons, we recorded miniature inhibitory postsynaptic currents (mIPSCs), which is considered to originate in single synapses, from postsynaptic neurons transfected with the control vector (*Figure 5A*, sh control) or the TEN2 knockdown vector (*Figure 5B*, sh Tenm2). Our analysis of mIPSCs revealed that the inter-event interval of mIPSC was prolonged significantly in the TEN2 knockdown neurons, although there was no significant difference in the mIPSC amplitude between the control and the knockdown neurons (*Figure 5C*). These results suggest that TEN2 knockdown reduces the frequency of mIPSC with no change in the single synaptic strength. This conclusion is in line with our *Figure 4* data showing that the number of matured inhibitory postsynapses (i.e. gephyrin puncta) is reduced in the TEN2 knockdown neurons while the expression of the GABA$_A$ receptor subunit α1 is intact.

## Interaction with MTs via EBs by two motifs in TEN2

Using live imaging of EB3 comets in neurons and quantitative analysis of immunostained inhibitory synapses, we propose that TEN2 is a potential MT recruiter at inhibitory synapses (*Figures 1D–H, and 2B–H*). To investigate the interaction between TEN2 and MTs, we conducted a GST-pulldown assay to examine the binding between TEN2 and EB. When GST-EB1 and GST-EB3 were used as bait, the amount of TEN2 pull-down was significantly higher than the control (*Figure 6A* and *Figure 6—figure supplement 1A–C*), indicating that TEN2 binds to EBs. To determine whether the SxφP and LxxPTPφ

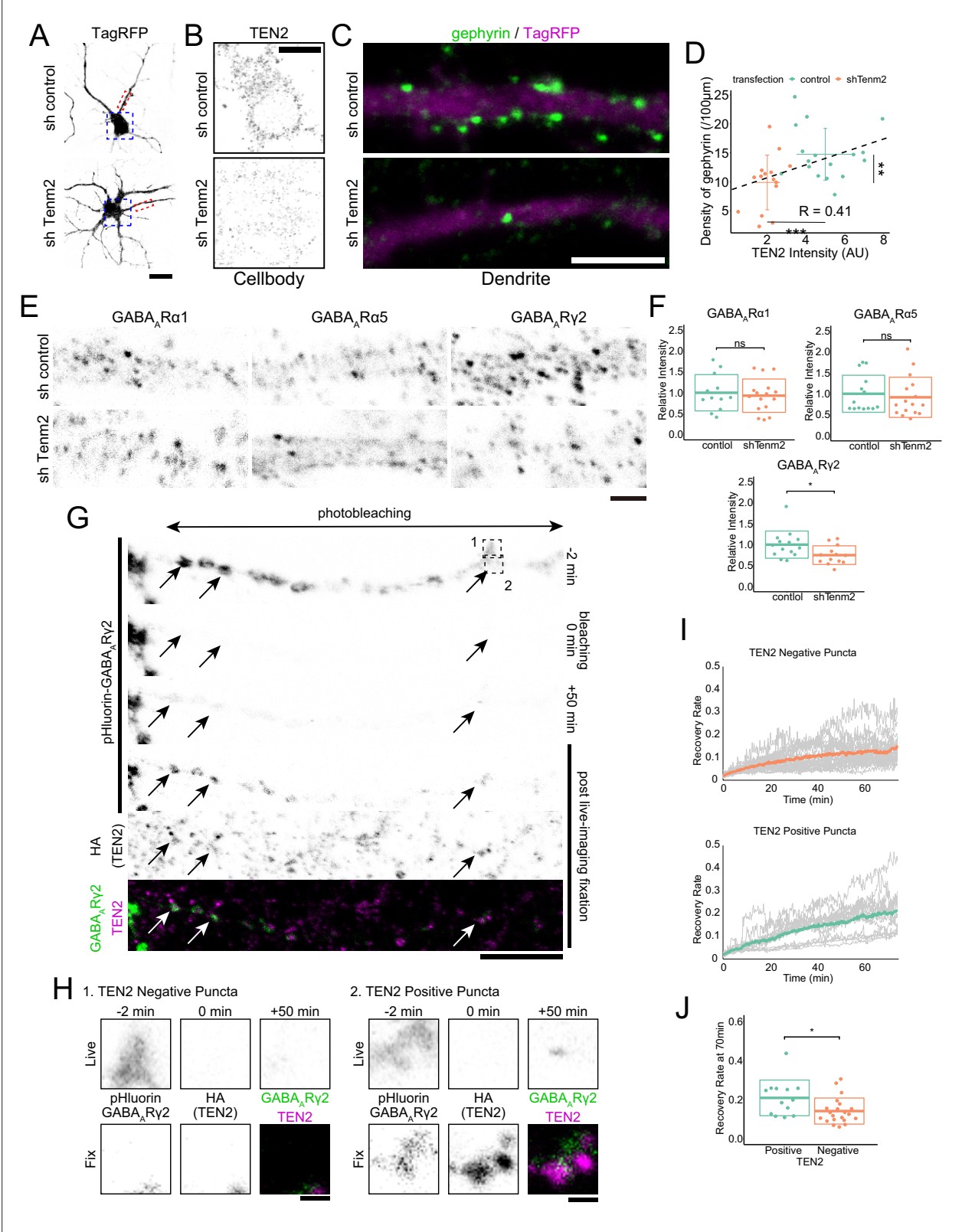

**Figure 4.** TEN2 provides a platform for the exocytosis of GABA$_A$ receptors at inhibitory postsynapses to mature synapses. (**A**) Images of neurons transfected with control or knockdown vector. The area boxed by the blue dash line is shown in (**B**), and the area boxed by the red dash line is shown enlarged in (**C**). Scale bar, 20 μm. (**B**) Magnified images of knockdown neurons immunostained with TEN2. Scale bar, 10 μm. (**C**) Magnified images of knockdown neurons immunostained with gephyrin. Gephyrin accumulation was reduced in TEN2 knockdown neurons. Scale bar, 5 μm. (**D**) A plot

*Figure 4 continued on next page*

*Figure 4 continued*

with crossbars (mean ± SD) of the relationship between TEN2 fluorescence intensity in the cell bodies and the density of gephyrin puncta per 100 μm dendrite. The black dashed line represents a linear approximation of the correlation between TEN2 intensity and gefillin density without distinguishing between control and knockdown neurons (*R*=0.42). It should be noted that transfection with a knockdown vector significantly reduced TEN2 intensity (p=9.9e-8) and gephyrin density (p=0.0058). n=17 for control neurons and n=15 for knockdown neurons. **p<0.01, ***p<0.001 by Welch's t-test. (**E**) Magnified images of knockdown neurons immunostained with GABAA receptors subunit α1, α5, and γ2. Only the γ2 receptor is downregulated in TEN2 knockdown neurons of these subunits. Scale bar, 2 μm. (**F**) Plots and cross bars (mean ± SD) quantifying the relative intensity of GABAA receptor subunits. The fluorescence intensities of receptors present in dendrites within 100 μm from the cell body were quantified comparatively. Mean ± SD were 1±0.43 and 0.93±0.40 for α1, 1±0.40 and 0.92±0.48 for α5, and 1±0.32 and 0.75±0.22 for γ2. Welch's t-test showed that α1 (p=0.67) and α5 (p=0.62) were not significantly different between control and TEN2 knockdown neurons. γ2 (p=0.027) was predominantly reduced in TEN2 knockdown neurons. n=12, 16, 14, 16, 14, and 13 neurons from three independent experiments. *p<0.05. (**G**) Time-lapse images showing FRAP assay and immunostaining of TEN2 in post-live-imaging fixation. Arrows indicate exocytosed GABAA receptors puncta in typical TEN2-positive positions. The area boxed by the dashed line is shown in (**H**). Scale bar, 10 μm. (**H**) Magnified images of FRAP assay. The pHluorin signal indicating surface expression of GABAARγ2 was observed 50 min after photobleaching in the TEN2 positive position, whereas the signal in the TEN2-negative position was very slight. Scale bar, 1 μm. (**I**) Statistical analysis showing signal recovery. Gray lines indicate the ratio of pHluorin-GABAARγ2 signal intensity after photobleaching to the intensity before photobleaching in individual puncta. Colored lines indicate mean values. (**J**) Plot and crossbars (mean ± SD) of recovery rate at 70 min after photobleaching. The recovery rate was significantly higher in TEN2 positive puncta (p=0.032). n=13 positive puncta and 21 negative puncta. *p<0.05 by Welch's t-test.

The online version of this article includes the following source data for figure 4:

**Source data 1.** 4 Excel sheets containing the numerical data used to generate the *Figure 4D, F,I and J*.

motifs of TEN2 are involved in binding to EBs, we assayed their co-localization in COS-7 cells using a fusion protein of partial domains and EGFP (*Figure 6B*). Endogenous EB1 is localized to the plus ends of MTs and observed as dynamic comets. However, this localization is lost upon cell fixation. Therefore, we overexpressed EB1 and localized it throughout MTs to detect protein-protein interactions (*Skube et al., 2010*). We first measured the correlation coefficient by overexpressing EB1-TagRFP and the chimeric proteins TEN2N-L, which consist of two EB1 binding motifs in the transmembrane and cytoplasmic domains of TEN2, in COS-7 cells. The results showed that TEN2N-L co-localized with EB1-TagRFP (*Figure 6C and D*). In contrast, TEN2TM, which has only a transmembrane domain, and TEN2N-L2mut, which has an amino acid mutation in the EB1 binding motif, did not co-localize with EB1 (*Figure 6C and E*). These findings suggest that the cytoplasmic domain of TEN2 interacts with MTs through binding to EBs via the SxφP and LxxPTPφ motifs.

## MTs need to be recruited near the cell membrane by TEN2 for inhibitory synapse formation

We aimed to investigate how the binding between TEN2 and EBs affects the function of TEN2 as a synaptic organizer. First, we examined the effect of the partial domains by live imaging of EB3 in neurons expressing them. We found that neurons expressing TEN2N-L showed kymographs with linear motion and little velocity change compared to control neurons expressing TEN2TM or TEN2N-L2mut (*Figure 7A and B*). Quantitatively, the run length and velocity of EB3 comets were significantly increased in TEN2N-L-expressing neurons (*Figure 7C and D*). However, there was no significant difference in comet duration between the three partial domains (*Figure 7E*). This suggests that TEN2N-L functions as a dominant-negative and that the interaction between endogenous TEN2 and EBs is lost.

Next, we investigated the effect of the interaction between TEN2 and EBs on inhibitory synaptic formation. We quantified the number of gephyrin puncta and GABA_A receptor γ2 subunit puncta in neurons expressing TEN2N-L and found that both were significantly reduced compared to TEN2TM (*Figure 7F–H*). This suggests that inhibitory synapse formation is decreased when TEN2N-L functions as a dominant-negative and inhibits the interaction between endogenous dendritic TEN2 and EBs. These findings suggest that TEN2 functions as a synapse organizer by recruiting MTs near the cell membrane of inhibitory postsynapse.

## Discussion

Since inhibitory synapses are formed directly on dendritic shafts, the distinction between inhibitory postsynaptic and non-synaptic membranes is determined solely by the accumulation of inhibitory synapse-specific components. Efficient accumulation during synapse formation requires a mechanism

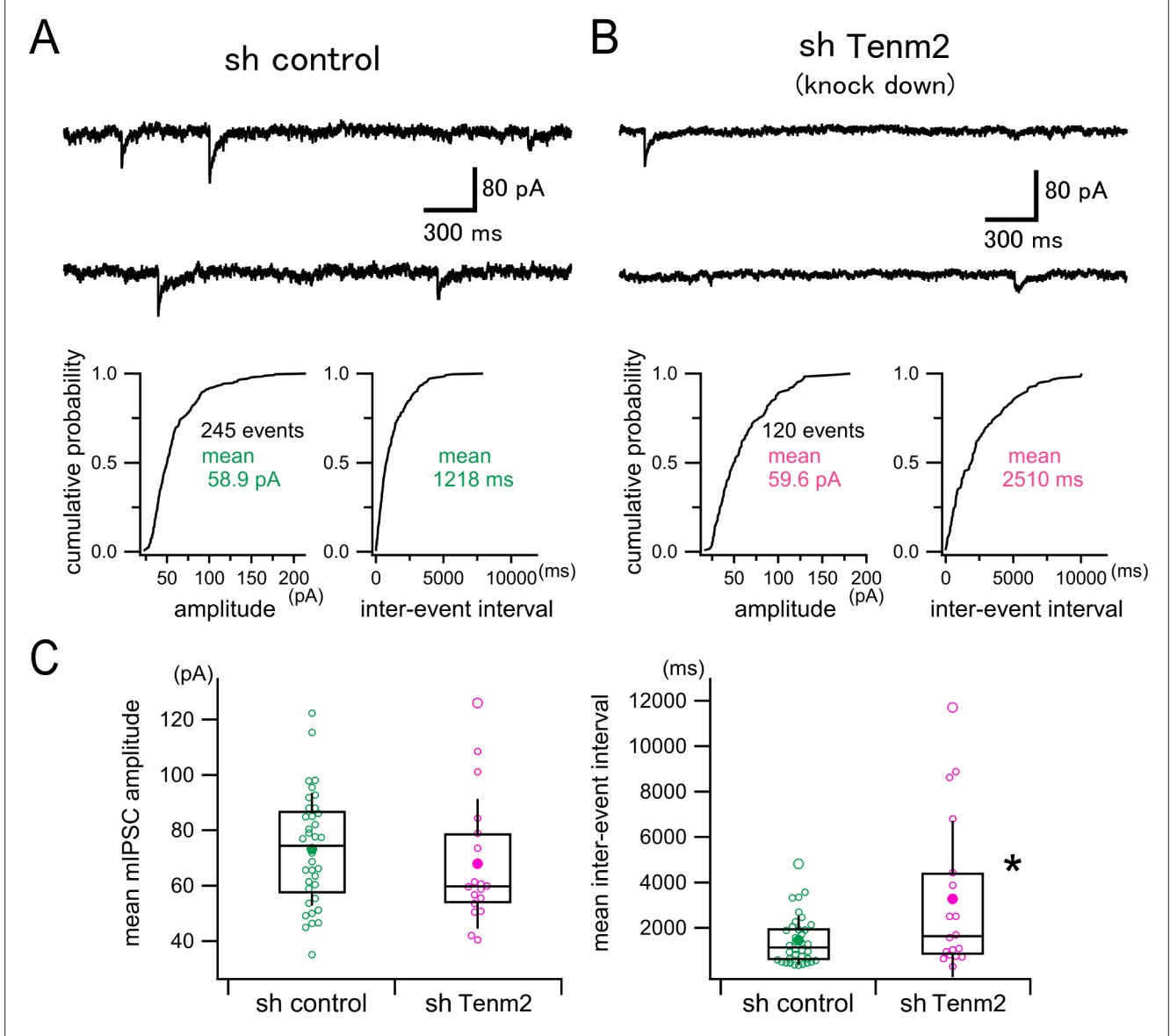

**Figure 5.** Effect of TEN2 knockdown on miniature inhibitory synaptic currents (mIPSCs) in cultured hippocampal neurons. (**A**) Upper panel shows a representative continuous 6 s trace (3 s traces in a row) of mIPSC recording in a control neuron (sh control). The lower panel shows the representative cumulative probability distributions of mIPSC amplitude (left) and inter-event interval (right) measured from a 300 s recording in this neuron. The single mean values of the amplitude and the interval were used to represent each neuron. (**B**) A representative example of a TEN2 knockdown neuron (sh Tenm2) is shown similarly to (**A**). (**C**) Box and whisker plots of mean mIPSC amplitudes (left) and mean inter-event intervals (right). Open circles correspond to individual data points, and the central horizontal lines and the boxes represent the median values and the interquartile ranges, respectively. Filled circles indicate the averaged values, and the error bars indicate one standard deviation above and below the values. TEN2 knockdown had no effect on mIPSC amplitude (sh control, 73.2±20.4 pA, n=36 neurons; sh Tenm2, 67.9±23.5 pA, n=18 neurons, Welch's t-test, p=0.424), but prolonged inter-event interval (i.e. reduced mIPSC frequency) significantly (sh control, 1455±1077ms, n=36 neurons; sh Tenm2, 3272±3444ms, n=18 neurons, Welch's t-test, p<0.0418). *p<0.05.

for preferential delivery of components to the synaptic region rather than the extrasynaptic region. We decided to examine the hypothesis that efficient transport can be achieved by recruiting MTs to the synapse. First, synapses were classified by the cytoskeleton, and three clusters were identified: MT-rich, actin-rich, and low in either. Additionally, EB3 comets were found to be dissipated at inhibitory synapses, suggesting the presence of MT recruiters in the synapse. Motif searches and quantitative analysis identified the synaptic organizers NLGN2 and TEN2 as candidates for MT recruiters. The present study has since focused on the analysis of TEN2 to confirm its function in accumulating

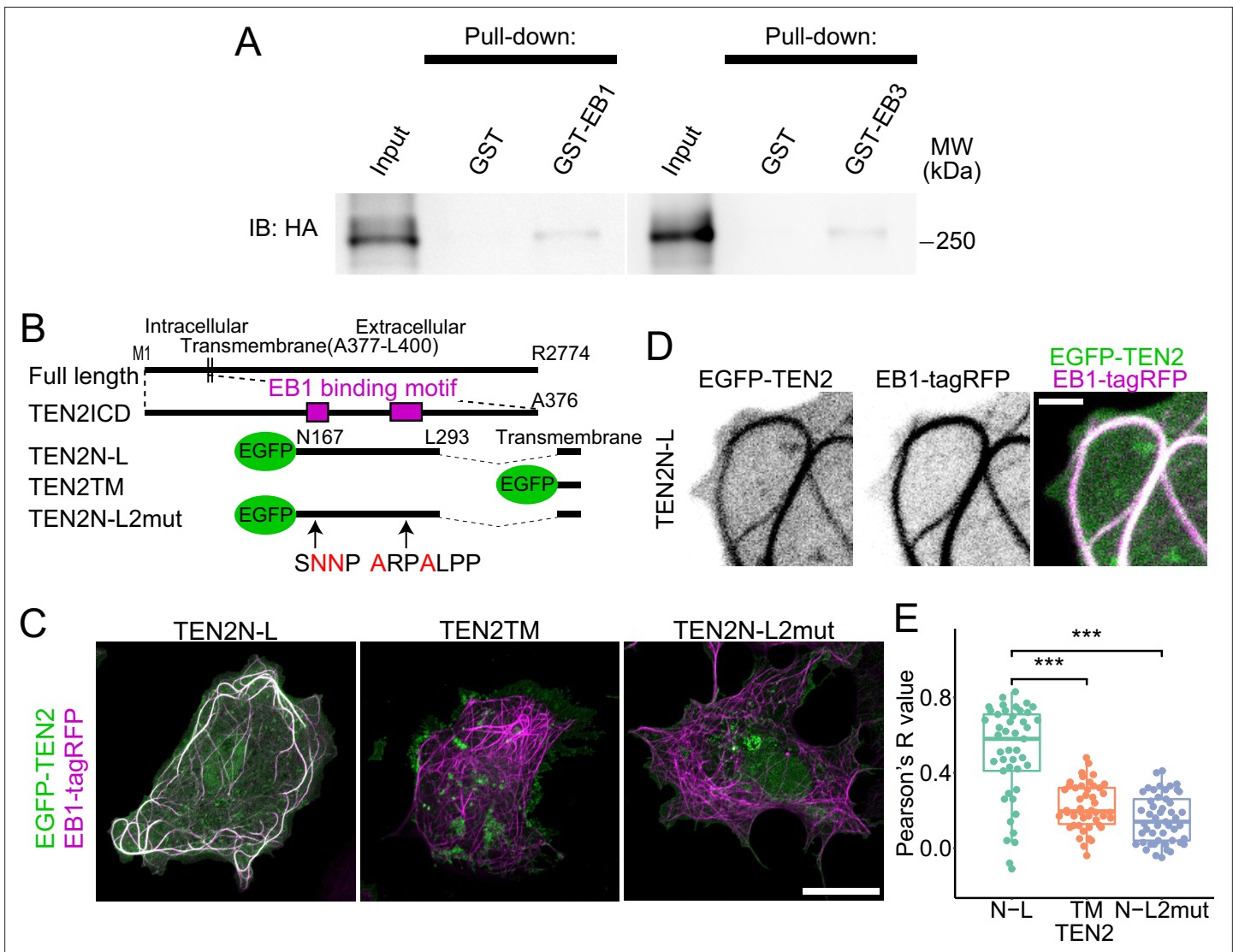

**Figure 6.** Interaction with MTs via EB1 by two motifs in TEN2. (**A**) Interaction between EB and TEN2 by pull-down assay. Pull-down assay was performed on brain lysate of TEN2-HA knock-in mice using GST-EB1/3 as bait, and both assays were positive for HA (TEN2) by Western blot. (**B**) Overview of the partial domain of TEN2N-L. TEN2N-L was designed to contain the two EB1 binding motifs detected by motif search. TEN2N-L2mut has amino acid mutations in two binding motifs. All proteins have transmembrane domains with predictable topogenic sequences. (**C**) Co-expression of each truncated mutant with EB1 in COS-7 cells. Cells with MTs patterns of over-expression of EB1 were observed. TEN2N-L colocalized well with EB1 compared to other partial domains, suggesting that TEN2 N-L interacts with EB1. Scale bar, 20 μm. (**D**) Highly magnified image of COS-7 cells expressing TEN2N-L. Scale bar, 2 μm. (**E**) Based on correlation coefficients, individual plots, and box plots show the quantitative analysis results of the colocalization index between each TEN2 and EB1. The median Pearson's correlation coefficient between TEN2N-L and EB1 was 0.58, which was significantly different from that of TEN2TM (0.195; p=1.3e-7), and TEN2N-L 2mut (0.14; p=2.9e-9) by Pairwise comparisons using Wilcoxon rank sum test after Kruskal-Wallis rank sum test (p=5.0e-11). The total number of cells observed was 46, 46, and 49, respectively. ***p<0.001.

The online version of this article includes the following source data and figure supplement(s) for figure 6:

**Source data 1.** 2 unprocessed full-size blot photographs showing western blotting of HA, as well as 2 photographs showing the region used in *Figure 6A* with dashed lines.

**Source data 2.** An Excel sheet containing the numerical data used to generate the *Figure 6E*.

**Figure supplement 1.** Interaction with MTs via EB1 by two motifs in TEN2.

**Figure supplement 1—source data 1.** 2 Unprocessed full-size gel photographs showing the results of GST pull-down used in *Figure 6—figure supplement 1A*, 2 unprocessed full-size blot photographs showing western blotting of HA used in *Figure 6—figure supplement 1B*, and 4 photographs showing the regions used in each figure with dashed lines.

**Figure supplement 1—source data 2.** An Excel sheet containing the numerical data used to generate the *Figure 6—figure supplement 1C*.

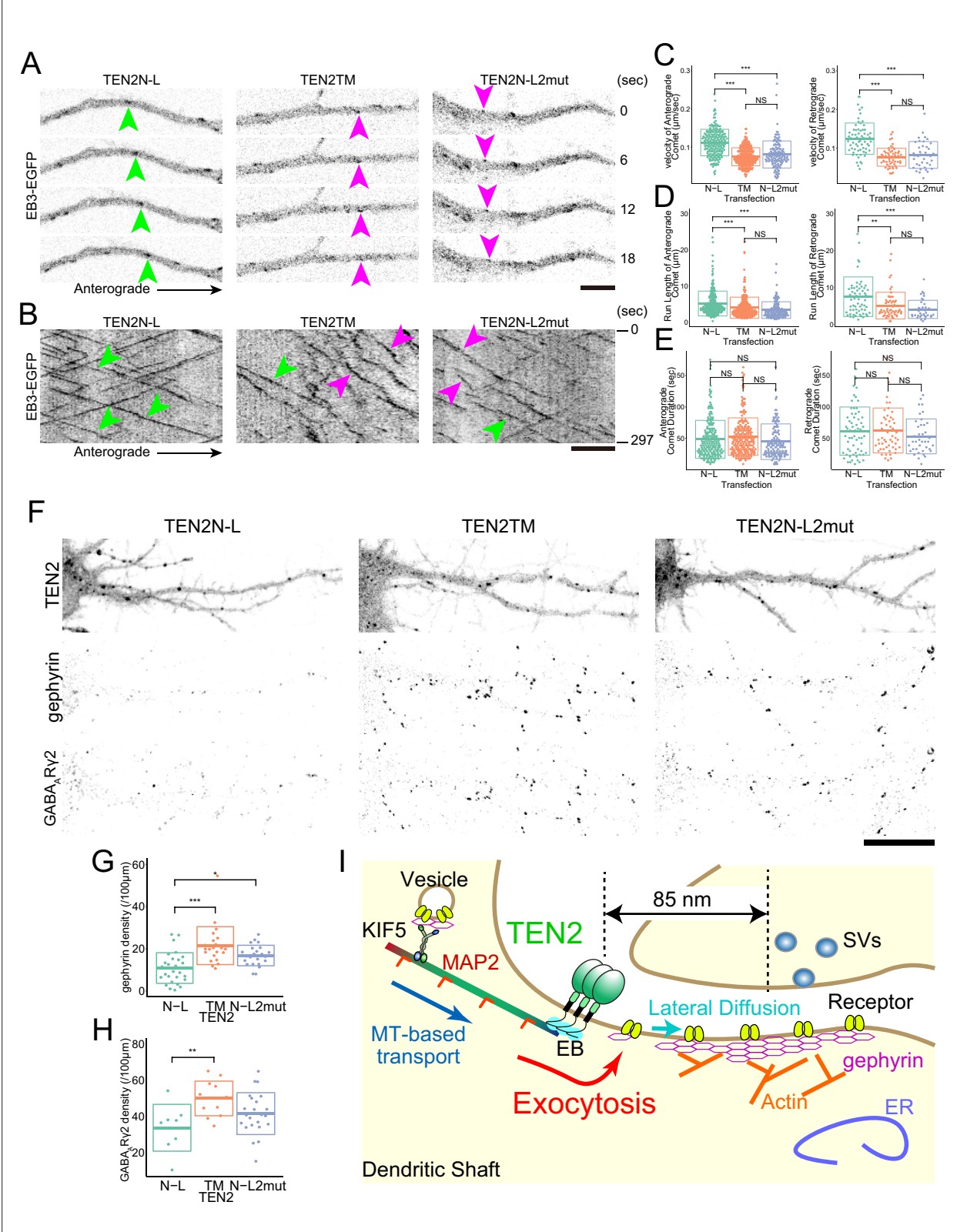

**Figure 7.** MTs need to be recruited near the cell membrane by TEN2 for inhibitory synapse formation. (**A**) Live imaging of EB3-EGFP in neurons expressing each partial domain. Due to the dominant-negative effect, the fast comet was observed in neurons expressing TEN2N-L (green arrowheads). In neurons expressing the other two domains, relatively slow comets (magenta arrowheads) were observed. Scale bar, 5 μm. (**B**) EB3-EGFP kymographs in the neurons expressing each partial domain, and a linear kymograph was observed in the TEN2N-L-expressing neurons due to the dominant-negative

*Figure 7 continued on next page*

*Figure 7 continued*

effect (green arrowheads). In neurons expressing the other two, undulation was observed in addition to linear kymograph (magenta arrowheads). Scale bar, 5 µm. (**C–E**) Statistical analysis of EB3-EGFP separately for anterograde and retrograde motion. The analysis revealed no significant difference in comet duration among the three partial domains (**E**). However, significant differences were observed in velocity (p<2e-7 for anterograde with TEN2TM, p<2e-7 for anterograde with TEN2N-L2mut, p<2e-7 for retrograde with TEN2TM, and p<2e-7 for retrograde with TEN2N-L2mut) and run length (p=1.5e-4 for anterograde with TEN2TM, p=3e-7 for anterograde with TEN2N-L2mut, p=2.7e-3 for retrograde with TEN2TM, and p=1.9e-4 for retrograde with TEN2N-L2mut), indicating a significant increase in TEN2N-L compared to the other two domains (**C and D**). For the anterograde motion, the statistical tests were based on a one-way ANOVA (p<2e-16 in C, p=1.4e-7 in D, and p=0.043 in E) followed by post hoc Tukey analysis. For the retrograde motion, the statistical tests were based on a one-way ANOVA (p=1.0e-13 in D, p=7.5e-5 in D, and p=0.147 in E) followed by post hoc Tukey analysis. The number of comets analyzed for the anterograde motion was as follows: TEN2N-L (n=208), TEN2TM (n=235), and TEN2N-L2mut (n=129). For the retrograde motion, the number of comets analyzed was as follows: TEN2N-L (n=66), TEN2TM (n=59), and TEN2N-L2mut (n=39). (**F**) Immuno-staining of gephyrin and GABAARγ2 subunits in neurons expressing each partial domain. Scale bar, 20 µm. (**G**) The density of gephyrin puncta in neurons expressing each partial domain. The density of gephyrin puncta was found to be significantly lower in neurons expressing TEN2N-L compared to those expressing TEN2TM (*P*=5.3e-6) and TEN2N-L2mut (*P*=0.013). The statistical tests were based on a one-way ANOVA (p=8.8e-6) followed by post hoc Tukey analysis. The sample sizes were as follows: TEN2N-L (n=28), TEN2TM (n=23), and TEN2N-L2mut (n=23). *p<0.05, ***p<0.001. (**H**) The density of GABAARγ2 puncta in neurons expressing each partial domain. The density of GABAARγ2 puncta was found to be significantly lower in neurons expressing TEN2N-L compared to those expressing TEN2TM (p=0.009). The statistical tests were based on a one-way ANOVA (p=0.011) followed by post hoc Tukey analysis. The sample sizes were as follows: TEN2N-L (n=8), TEN2TM (n=12), and TEN2N-L2mut (n=23). **p<0.01. (**I**) A working model derived from this study. The interaction of TEN2 and dynamic MTs provides a platform for exocytosis and allows proper transport of components of the inhibitory postsynapse.

The online version of this article includes the following source data and figure supplement(s) for figure 7:

**Source data 1.** 5 Excel sheets containing the numerical data used to generate the *Figure 7C–E, G and H*.

**Figure supplement 1.** MTs need to be recruited near the cell membrane by TEN2 for inhibitory synapse formation (**A**) Confocal imaging of gephyrin accumulation and MAP2 in neurons expressing each TEN2.

**Figure supplement 1—source data 1.** 2 Excel sheets containing the numerical data used to generate the *Figure 7—figure supplement 1B and C*.

---

inhibitory postsynaptic components. Super-resolution microscopy, knockdown experiments, and FRAP assays were performed, ultimately demonstrating that TEN2 functions as an organizer by providing a platform for the exocytosis of GABA_A receptors. Next, we investigated whether an TEN2-MTs interaction underlies this function. Through pull-down assays and observation of co-localization between TEN2 partial domain and EB, we found that TEN2 interacts with MTs via EBs. We also found that this partial domain functions as a dominant-negative in neurons, inhibiting the binding of endogenous TEN2 to EBs and suppressing inhibitory synapse formation. Based on these results, we conclude that TEN2 interacts with MTs via EBs at inhibitory postsynapses and induces the exocytosis of GABA_A receptors (*Figure 7I*).

## Relationship between TEN2 and exocytosis

The FRAP assay has shown that TEN2 regulates protein distribution by providing a site for GABA_A receptors exocytosis (*Figure 4G–J*). Previous studies have reported that exocytosis of glutamate and GABA_A receptors occurs in different locations (*Gu et al., 2016*). Our study has newly identified the site of exocytosis of GABA_A receptors. However, the detailed mechanisms of this process still require further validation. This is because TEN2N-L alone cannot induce exocytosis simply by interacting with EBs (*Figure 7F*). These results suggest that TEN2 has additional domains and undergoes conformational changes necessary for exocytosis. The intracellular domain of TEN2, which is over 360 amino acids long, mainly consists of intrinsically disordered regions with no secondary structure, and its flexibility and structural diversity may allow for a variety of functions, including signal transduction.

The antibodies used in this study provide insight into the question of how TEN2 interacts with EBs. Antibodies targeting the vicinity of the EB-binding motif were successfully generated, but only moderate co-staining was observed when stained with anti-HA antibodies (*Figure 3—figure supplement 1E–H*). This result suggests that, on the one hand, the antibodies may nonspecifically recognize other molecules. On the other hand, assuming that the antibody accurately recognizes only TEN2, it may preferentially recognize specific structures. For instance, an epitope may be open around the nucleus and at synapses but structurally folded or masked at other localities due to interactions with other molecules. It is also noteworthy that only small amounts of TEN2 can bind to EBs, as seen in the pull-down assay. Furthermore, the fact that EB3 live imaging shows EB dispersing at the synapse, despite TEN2 also localizing extrasynaptically, suggests that only specific structures interact with EBs.

Such conformational changes favor the EB-binding motifs in the vicinity of the epitope to be open and functional at the synapse. The mechanism by which the EB-binding motif opens and closes has also been demonstrated in the ER membrane protein STIM1 (*Chang et al., 2018*). Therefore, to clarify the structural diversity of TEN2, differences in binding molecules, and localization, it will be essential to generate other structure-recognition antibodies and analyze the structure of complexes with interacting proteins.

We have demonstrated that TEN2 provides a platform for the exocytosis, but we have not yet determined how cargo is released by kinesin and exocytosed. Potential mechanisms for this process include phosphorylation of kinesins, post-translational modifications of tubulin (tubulin code; *Janke and Magiera, 2020*), and MAP-mediated effects on kinesins and MTs (MAPs code; *Aiken and Holzbaur, 2021*; *Monroy et al., 2020*). However, phosphorylation-based regulation of kinesins poses a challenge for synaptic formation since each protein kinase has a unique distribution pattern, making the problem of how to distribute cargo equivalent to the problem of how to distribute kinase (*Ichinose et al., 2015*; *Ichinose et al., 2019*). In contrast, regulatory mechanisms involving tubulin status and MAPs-mediated effects only require a single step of interaction between adhesion molecules and MTs, which may simplify the process of distributing cargo.

For example, recent studies have shown that competition between EB1 and the kinesin KIF1A reduces the affinity between kinesin and MTs, leading to cargo release (*Guedes-Dias et al., 2019*; *Qu et al., 2019*). Applying this to the transport of GABA$_A$ receptors, if TEN2 recruits the plus end of MTs, competition between KIF5 and EB may occur, increasing the probability of KIF5 detachment from MTs. However, it is important to avoid conflicts between 'tubulin code' and 'MAPs code' because KIF5 does not readily detach from GTP MTs at the plus end of MTs (*Nakata et al., 2011*). The competition between EB and kinesin may be specific to KIF1A, or there may be signaling within TEN2 that counteracts one of the conflicting codes when both tubulin and MAPs codes are present simultaneously. Thus, detailed domain analysis of the TEN2 side and elucidation of the cargo release mechanism on the kinesin side are essential for explaining the detailed mechanisms in the future.

## Undulations of EB comets on kymographs

In experiments with partial domains, it was observed that TEN2N-L acted as a dominant-negative. Kymographs of neurons expressing TEN2TM or TEN2N-L2mut as controls exhibited undulations, while those expressing TEN2N-L showed counteraction of undulations (*Figure 7B*). The cause of the undulations is likely due to weaker capture compared to synapses and may reflect immature regions that will develop into synapses in the future. However, further live imaging of developing synapses over several days is necessary to test this possibility.

Our findings suggest that TEN2N-L can counteract undulations in the kymograph by activating MT elongation through the increased concentration of EBs recruited by TEN2N-L, which is supported by the biased localization of MTs toward the submembrane (*Figure 7—figure supplement 1A–C*). Nonetheless, our results also indicate that although endogenous TEN2 recruits EBs, the comet must be disrupted to counteract MT elongation. It is possible that the binding of endogenous TEN2 to its partner molecules as an adhesion molecule antagonizes MT elongation in a kinetic manner. To confirm this hypothesis, measuring the magnitude of the kinetic energy of each molecule in the neuron may be necessary.

## Analysis of the relationship between synaptic organizers and cytoskeletal molecules

We investigated the function of TEN2 on the dynamics of GABA$_A$ receptors by combining live imaging of EB and GABA$_A$ receptors with immunostaining of fixed cells. Based on previous studies, we inferred that synapses that are rich in MTs exhibit active transport of components, while synapses that are rich in actin have stable receptors. Synapses that have fewer of both types of cytoskeletal elements exhibit active lateral diffusion of components (*Figure 2—figure supplement 1A*; *Charrier et al., 2006*; *Fuhrmann et al., 2002*; *Giesemann et al., 2003*; *Gu et al., 2016*; *Kittler et al., 2000*; *Labonté et al., 2014*; *Nakajima et al., 2012*; *Twelvetrees et al., 2010*). However, it is important to note that all related proteins has dynamic properties. In our study, we observed exocytosis of GABA$_A$ receptors by labeling them with pHluorin. However, there are limitations to observing intracellular transport and lateral diffusion using this labeling method. Single-molecule imaging with Qdot is superior to pHluorin

for observing lateral diffusion (*Dahan et al., 2003*). Thus, to fully understand the dynamic relationship between the cytoskeleton and synaptogenesis, it is necessary to simultaneously visualize GABA$_A$ receptors with pHluorin and Qdot while live imaging EBs and Lifeact with fluorescent proteins. However, this approach presents significant challenges, including the mismatch in the required temporal resolution for observing each molecule and the demanding scan speed and spectral separation.

In addition, it is often problematic that overexpression systems do not accurately reflect the localization of endogenous proteins, as they may undergo protein redistribution and be retained in intracellular organelles such as the endoplasmic reticulum and proteolytic system. To investigate the gain-of-function effect of TEN2, we attempted to replicate the synapse formation induction assay performed in non-neuronal cells (*Sando et al., 2019*) by overexpressing full-length TEN2 in neurons. However, we were unsuccessful as the localization pattern of the overexpressed TEN2 differed from that of the endogenous TEN2. As a result, our research primarily focuses on loss-of-function studies. Nevertheless, it is important to explore aspects of gain-of-function using alternative methods in the future, because they may hold potential for treating psychiatric disorders that arise from excitatory/inhibitory imbalance.

Since the issue of overexpression is not limited to TEN2, we selected cells with minimal observable expression and performed immunostaining after live imaging to avoid artifacts caused by unnecessary overexpression (*Figures 1D and 4G*). In the future, labeling endogenous proteins for live imaging will be necessary to overcome this problem. Currently, various knock-in techniques for multiple fluorescent proteins and the development of bright and stable fluorescent proteins that can mimic endogenous protein levels are being investigated (*Droogers et al., 2022*; *Hirano et al., 2022*).

Furthermore, although we focused only on TEN2 in this study, it is essential to distinguish other synaptic organizers to fully understand the molecular mechanisms underlying synaptic development and function. For example, there are other molecules with EB-binding motifs besides TEN2, and the functions of their respective intracellular domains are still largely unknown (*Figure 2H*). Some of these molecules may be functionally homologous, while others may not be. On the other hand, many reports suggest that each organizer selects its binding partner based on differences in extracellular domains, contributing to the formation of dense neural circuits. Whether differences in organizers simply indicate differences in neural circuits or also reflect differences in intracellular signaling is a future focus.

Thus, to fully understand the input from individual interneurons and their signaling modalities at postsynapses, it will be necessary to perform simultaneous multicolor observations of these organizers, cytoskeletal molecules, and GABA$_A$ receptors. This approach will allow us to distinguish between the contributions of different molecules to the formation of neural circuits and the modulation of intracellular signaling, providing important insights into the organization and function of synapses. In addition, this approach may enable us to identify specific molecular mechanisms underlying psychiatric disorders caused by certain molecular abnormalities, including E/I imbalances. By understanding the roles of different molecules in the development of psychiatric disorders, we may be able to develop more effective treatments for these debilitating conditions.

# Materials and methods

## Key resources table

| Reagent type (species) or resource | Designation | Source or reference | Identifiers | Additional information |
|---|---|---|---|---|
| Strain, strain background (*Mus musculus*) | C57BL/6 J JAX mice | Charles River | Cat# JAX:000664, RRID:IMSR_ JAX:000664 | |
| Strain, strain background (*Mm*) | ICR | Japan SLC | Cat# 5462094, RRID:MGI:5462094 | |
| Strain, strain background (*Mm*) | ICR | Charles River | Cat# CRL:022, RRID:IMSR_CRL:022 | |
| Strain, strain background (*Escherichia coli*) | 5-alpha Competent | New England Biolabs | Cat# C2987 | |
| Strain, strain background (*Ec*) | BL21(DE3) Competent Cells | Agilent | Cat# 200131 | |

*Continued on next page*

*Continued*

| Reagent type (species) or resource | Designation | Source or reference | Identifiers | Additional information |
|---|---|---|---|---|
| Cell line (*Chlorocebus sabaeus*) | COS-7 cells | RIKEN Cell Bank | Cat# RCB0539, RRID:CVCL_0224 | |
| Antibody | Rabbit polyclonal anti-TEN2 Cytoplasmic | This study | N/A | 0.2–0.8 µg/mL |
| Antibody | Rabbit polyclonal GABRA1 antibody | Proteintech | Cat# 12410–1-AP, RRID:AB_2108692 | 1:1000 |
| Antibody | Rabbit polyclonal Anti-GABA-A receptor alpha5 | Synaptic Systems | Cat# 224 503, RRID:AB_2619944 | 1:5000 |
| Antibody | Rabbit polyclonal Anti-GABA-A receptor gamma2 | Synaptic Systems | Cat# 224 003, RRID:AB_2263066 | 1:2000 |
| Antibody | Mouse monoclonal anti-gephyrin (mAb7a) | Synaptic Systems | Cat# 147011, RRID:AB_887717 | 1:2000 |
| Antibody | Chicken polyclonal Anti-Bassoon | Synaptic Systems | Cat# 141 016, RRID:AB_2661779 | 1:2000 |
| Antibody | Mouse monoclonal Anti-VGAT(117G4) | Synaptic Systems | Cat# 131 011, RRID:AB_887872 | 1:5000 |
| Antibody | Rabbit polyclonal Anti-Neuroligin 2 | Synaptic Systems | Cat# 129 203, RRID:AB_993014 | 1:2000 |
| Antibody | Mouse monoclonal anti-PSD95 (7E3) | Cell Signaling Technology | Cat# 36233, RRID:AB_2721262 | 1:1000 |
| Antibody | Rabbit monoclonal anti-HA-tag (C29F4) | Cell Signaling Technology | Cat# 3724, RRID:AB_1549585 | 1:1000 (IF, WB) |
| Antibody | Rabbit polyclonal Anti-IGSF9B | Merck | Cat# HPA010802, RRID:AB_1079194 | 1:1000 |
| Antibody | Chicken polyclonal anti-MAP2 | Novus | Cat# NB300-213, RRID:AB_2138178 | 1:50000 |
| Antibody | Donkey polyclonal Anti-Mouse IgG (Alexa Fluor 405) | abcam | Cat# ab175658, RRID:AB_2687445 | 1:1000 |
| Antibody | Donkey polyclonal Anti-Mouse IgG (H+L), Alexa Fluor 488 | Jackson ImmunoResearch Labs | Cat# 715-546-151, RRID:AB_2340850 | 1:2000 |
| Antibody | Donkey polyclonal Anti-Mouse IgG (H+L), Rhodamine Red-X | Jackson ImmunoResearch Labs | Cat# 715-296-151, RRID:AB_2340835 | 1:2000 |
| Antibody | Donkey polyclonal Anti-Mouse IgG (H+L), Alexa Fluor 647 | Jackson ImmunoResearch Labs | Cat# 715-606-151, RRID:AB_2340866 | 1:2000 |
| Antibody | Donkey polyclonal Anti-Rabbit IgG (H+L), DyLight 405 | Jackson ImmunoResearch Labs | Cat# 711-475-152, RRID:AB_2340616 | 1:1000 |
| Antibody | Donkey polyclonal Anti-Rabbit IgG (H+L), Alexa Fluor 488 | Jackson ImmunoResearch Labs | Cat# 711-546-152, RRID:AB_2340619 | 1:2000 |
| Antibody | Donkey polyclonal Anti-Rabbit IgG (H+L), CF568 | Biotium | Cat# 20098–1, RRID:AB_10853318 | 1:2000 |
| Antibody | Donkey polyclonal Anti-Chicken IgY (IgG) (H+L), Alexa Fluor 647 | Jackson ImmunoResearch Labs | Cat# 703-605-155, RRID:AB_2340379 | 1:2000 |
| Antibody | Donkey polyclonal Anti- Rabbit IgG (H+L), HRP | Jackson ImmunoResearch Labs | Cat# 711-036-152, RRID:AB_2340590 | 1:20000 (WB) |
| Recombinant DNA reagent | guide RNA for knock-in | IDT | | 5'- GACAGAATGAGATGGGAAAG -3' |

*Continued on next page*

*Continued*

| Reagent type (species) or resource | Designation | Source or reference | Identifiers | Additional information |
|---|---|---|---|---|
| Recombinant DNA reagent | ssODN for knock-in | IDT | | 5'-ACAGTAGCAGCAACATCCAGTT CTTAAGACAGAATGAGATGGGAAA GAGATACCCATACGATGTACCTGAC TATGCGGGCTATCCCTATGACGTC CCGGACTATGCAGGATCCTATCCT TATGACGTTCCAGATTACGCTGTTT AACAAAATAACCTGCTGCCACCTC TTCTCTGGGTGGCTCAGCAGGAGCAACT-3' |
| Recombinant DNA reagent | *Homo sapiens TENM2* cDNA | KAZUSA | NCBI AB032953 | TEN2 |
| Recombinant DNA reagent | *Mm Tenm2* cDNA | RIKEN | NCBI AK031198 | TEN2 |
| Recombinant DNA reagent | *Hs MAPRE1* cDNA | KAZUSA | NCBI AB463888 | EB1 |
| Recombinant DNA reagent | *Hs MAPRE3* cDNA | Eurofins Genomics | | EB3 gene synthesis |
| Recombinant DNA reagent | pHluorin-GABA$_A$Rγ2 | Addgene | plasmid # 49170 RRID:Addgene_49170 | *Jacob et al., 2005* |
| Recombinant DNA reagent | pBAsi-mU6 DNA | Takara Bio | Cat# 3222 | |
| Recombinant DNA reagent | Top strand of oligonucleotide cassette for control shRNA | Eurofins Genomics | | 5'-GATCCGGCCTAAGGTT AAGTCGC CCTCGCTCGAGCGAGGGCGACT TAACCTTAGGTTTTTGA –3' |
| Recombinant DNA reagent | Bottom strand of oligonucleotide cassette for control shRNA | Eurofins Genomics | | 5'-AGCTTCAAAAACCTAA GGTTAA GTCGCCCTCGCTCGAGCGAGGG CGACTTAACCTTAGGCCG –3' |
| Recombinant DNA reagent | Top strand of oligonucleotide cassette for Tenm2 shRNA | Eurofins Genomics | | 5'-GATCCGGGCCAGGTTTG ATTATACCTATCTCGAGATA GGTATAATCAAACCTGGCTT TTTGA –3' |
| Recombinant DNA reagent | Bottom strand of oligonucleotide cassette for Tenm2 shRNA | Eurofins Genomics | | 5'-AGCTTCAAAAAGCCAGGTTT GATTATACCTATCTCGAGATAGG TATAATCAAACCTGGCCCG –3' |
| Recombinant DNA reagent | Top strand of oligonucleotide cassette for LifeAct | Eurofins Genomics | | 5'-CTAGCATGGGCGTGGCCGA CCTGATCAAGAAGTTCGAATCG ATAAGCAAGGAAGAGGGC –3' |
| Recombinant DNA reagent | Bottom strand of oligonucleotide cassette for LifeAct | Eurofins Genomics | | 5'-GATCGCCCTCTTCCTTGCTT ATCGATTCGAACTTCTTGATCA GGTCGGCCACGCCCATG –3' |
| Peptide, recombinant protein | synthetic peptide | Eurofins Genomics | | CSNTSHQIMDTNPDE |
| Peptide, recombinant protein | synthetic peptide | GenScript | | CQMPLLDSNTSHQIMD TNPDEEFSPNS |
| Commercial assay or kit | FlexAble CoraLite 488 Antibody Labeling Kit for Rabbit IgG | Proteintech | Cat# KFA001 | |
| Commercial assay or kit | FlexAble CoraLite Plus 555 Antibody Labeling Kit for Rabbit IgG | Proteintech | Cat# KFA002 | |
| Commercial assay or kit | FlexAble CoraLite Plus 647 Antibody Labeling Kit for Rabbit IgG | Proteintech | Cat# KFA003 | |
| Commercial assay or kit | Zenon Mouse IgG1 Labeling Kits Alexa Fluor 405 | Thermo Fisher Scientific | Cat# Z25013 | |

Continued

| Reagent type (species) or resource | Designation | Source or reference | Identifiers | Additional information |
|---|---|---|---|---|
| Commercial assay or kit | Zenon Mouse IgG1 Labeling Kits Alexa Fluor 594 | Thermo Fisher Scientific | Cat# Z25007 | |
| Commercial assay or kit | Duolink In Situ PLA Probe Anti-Mouse PLUS | Merck | Cat# DUO92001 | |
| Commercial assay or kit | Duolink In Situ PLA Probe Anti-Rabbit MINUS | Merck | Cat# DUO92005 | |
| Commercial assay or kit | Duolink In Situ Detection Reagents Green | Merck | Cat# DUO92014 | |
| Commercial assay or kit | High-Efficiency $Ca^{2+}$ Phosphate Transfection Kit | Takara Bio | Cat# 631312 | |
| Chemical compound, drug | Lipofectamine 2000 Transfection Reagent | Thermo Fisher Scientific | Cat# 11668030 | |
| Chemical compound, drug | Alexa Fluor 555 Phalloidin | Thermo Fisher Scientific | Cat# A34055 | |
| Chemical compound, drug | Can Get Signal Solution | Toyobo | Cat# NKB-101 | |
| Chemical compound, drug | Immunostar Zeta | FUJIFILM Wako | Cat# 291–72401 | |
| Chemical compound, drug | polyethylenimine solution | Merck | Cat# P3143 | |
| Chemical compound, drug | BioCoat poly-D-lysine | Corning | Cat# 354210 | |
| Chemical compound, drug | MEM, no glutamine | Thermo Fisher Scientific | Cat# 11090081 | |
| Chemical compound, drug | GlutaMAX | Thermo Fisher Scientific | Cat# 35050061 | |
| Chemical compound, drug | B27 Plus Plus Supplement (50 X) | Thermo Fisher Scientific | Cat# A3582801 | |
| Chemical compound, drug | SulfoLink Coupling Resin | Thermo Fisher Scientific | Cat# 20401 | |
| Chemical compound, drug | Glutathione Sepharose 4B | Cytiva | Cat# 17075601 | |
| Software, algorithm | Fiji | NIH | https://fiji.sc | |
| Software, algorithm | KymoResliceWide | Eugene Katrukha | https://imagej.net/KymoResliceWide | |
| Software, algorithm | KymographClear 2.0 a | Erwin Peterman's group | https://sites.google.com/site/kymographanalysis/ | Mangeol et al., 2016 |
| Software, algorithm | R | R Core Team | https://www.r-project.org | |
| Software, algorithm | pCLAMP | Molecular Devices | https://www.moleculardevices.com/ | |
| Software, algorithm | Igor Pro 8 | Wavemetrics | https://www.wavemetrics.com/software/igor-pro-8 | |
| Software, algorithm | NeuroMatic | ThinkRandom | http://www.neuromatic.thinkrandom.com/ | Rothman and Silver, 2018 |

## CRISPR/Cas9-mediated knock-in of the 3×HA tag into Tenm2 gene

The Tenm2 3×HA tag knock-in mice were generated by using a CRISPR/Cas9 genome-editing technology onto pronuclear stage embryos (*Doudna and Charpentier, 2014*). In brief, female C57BL/6 J JAX mice (Charles River; IMSR Cat# JAX:000664, RRID:IMSR_JAX:000664) were super-ovulated by intraperitoneal injection of 7.5 units of pregnant mare's serum gonadotropin (PMSG; ASKA Pharmaceutical), followed by 7.5 units of human chorionic gonadotropin (hCG; ASKA Pharmaceutical) 48 hr later. Fifteen hours after the hCG injection, super-ovulated female mice were euthanized via cervical dislocation, and unfertilized oocytes isolated from the female mice were subjected to in vitro fertilization with freshly isolated spermatozoa from euthanized C57BL/6 J JAX male mice, as previously described (*Kaneko et al., 2018*). Introduction of Cas9 protein, guide RNA, and single strand oligodeoxynucleotide (ssODN) into pronuclear stage embryos was carried out using the TAKE method

(*Kaneko, 2017*). Cas9 protein, guide RNA, and ssODN were purchased from IDT (Integrated DNA Technologies). Mixture of crRNA and tracrRNA was used as guide RNA. Guide RNA and ssODN were designed to insert 3×HA tag sequences just upstream from the stop codon of the Tenm2 gene of the C57BL/6 mouse (guide RNA: 5'- GACAGAATGAGATGGGAAAG-3', ssODN: 5'-ACAGTAGCAGCA ACATCCAGTTCTTAAGACAGAATGAGATGGGAAAGAGATACCCATACGATGTACCTGACTATGC GGGCTATCCCTATGACGTCCCGGACTATGCAGGATCCTATCCTTATGACGTTCCAGATTACGCTGTTT AACAAAATAACCTGCTGCCACCTCTTCTCTGGGTGGCTCAGCAGGAGCAACT-3', where 3×HA tag sequences are underlined). The CRISPR/Cas9 solution contained 50 ng/µL Cas9 protein, 50 ng/µL crRNA, 50 ng/µL tracrRNA, and 100 ng/µL ssODN in Opti-MEM (Thermo Fisher Scientific). Super electroporator NEPA21 (NEPA GENE) was used to introduce Cas9 protein, guide RNA, and ssODN into embryos. The poring pulse was set to voltage: 225 V, pulse length: 2.0ms, pulse interval: 50ms, number of pulses: 4, decay rate: 10%, polarity: +. The transfer pulse was set to a voltage: 20 V, pulse length: 50ms, pulse interval: 50ms, number of pulse: 5, decay rate: 40%, Polarity: +/-. The CRISPR/Cas9 solution (45 µL) was filled between metal plates of 5 mm gap electrodes on a glass slide (CUY505P5, NEPA GENE). The embryos placed in line between the electrodes were then discharged. The embryos were then cultured in HTF at 37 °C in 5% $CO_2$/95% air. On the next day, two-cell embryos were transferred into the oviduct ampulla (40–48 embryos per oviduct) of pseudopregnant ICR (Japan SLC; MGI Cat# 5462094, RRID:MGI:5462094) females. All mice generated were genotyped by PCR amplification of genomic DNA isolated from the tip of tail, followed by sequencing. Sequence of the primers used for genotyping were as follows; Tenm2ex29+564 F; CAAGGAGCAGCAGAAAGCCAG; Tenm2ex29+871 R; TAAAGCAGCCCGGCCTCAGTG. The resulting PCR product was cut by BamHI, and the expected size was 308 bp for wild-type and 254 bp and 147 bp for Tenm2 3×HA tag knock-in mice. Mice were backcrossed with wild-type C57BL/6 J at least four times, and at least one of them was with a wild-type male to replace the Y chromosome. Mice were kept in a specific pathogens free environment according to the institutional guidelines of Gunma University. All mice have been geno-typed by PCR. These experiments have passed a rigorous ethical review and have been approved by Gunma University for animal experiments (approval number: 20–061) and genetic recombination experiments (approval number: 21–042).

## Cell culture

COS-7 cells (RCB Cat# RCB0539, RRID: CVCL_0224) were obtained from the RIKEN Cell Bank. It was established by Yakov Gluzman and deposited at RIKEN by Kazuo Todokoro. At RIKEN, Authentic Kit (AGC Techno Glass) was used to confirm the presence of African green monkey-derived cells. This confirmation was based on the band patterns observed after the electrophoresis of cell extracts reacted with five enzymes: LD, NP, G6PD, AST, and MPI. Furthermore, the identification of the animal species was performed by detecting specific sequences of mitochondrial DNA present in the cells using 18SrRNA as an internal control (Fw: CGGGGAATYAGGGTTCGATTC, Rv: GCCTGCTGCCTT CCTTKGATG) and species-specific primers (Fw: AAATCAAGGCATAGCTTAACGC, Rv: GGCCAACT ATGGTAGTTATGGT, Gene Bank: AY863426.1 as a reference) through PCR analysis. The cells were introduced to our laboratory after authentication (Lot #32). Regular DNA staining tests have confirmed the absence of mycoplasma contamination, and the results have consistently been negative. Cells were cultured in High-Glucose DMEM (FUJIFILM Wako) supplemented with 10% fetal bovine serum (BioWest) in T75 flasks (Thermo Fisher Scientific) at not more than 80% confluency. Hippocampi were dissected from ICR mice (Charles River; IMSR Cat# CRL:022, RRID:IMSR_CRL:022) or knock-in mice on embryonic day 16 (E16). No gender determination was done, and three or more embryos were used. The hippocampi were digested with 0.25% trypsin (Thermo Fisher Scientific) in HBSS (FUJIFILM Wako) for 15 min at 37 °C. Dissociated hippocampal cells were seeded at a density of $2×10^4$ cells per well on Lab-Tek II 8-well chamber coverglasses (Thermo Fisher Scientific) or 8 well chamber cover (Matsu-nami Glass) coated with 0.04% polyethylenimine (Merck) and BioCoat poly-D-lysine (Corning), or at $8×10^4$ cells/cm$^2$ on φ12mm circular coverslips (Paul Marienfeld) for electrophysiological experiments. Circular coverslips were sonicated in 1 M KOH for 15 min, washed with ultrapure water, and UV-steril-ized before use. All primary cells were cultured in MEM (Thermo Fisher Scientific) supplemented with 1 mM pyruvate (Thermo Fisher Scientific), 0.6% glucose, 2 mM GlutaMAX (Thermo Fisher Scientific), 2% B27 Plus (Thermo Fisher Scientific), and 100 U/mL Penicillin-Streptomycin (Thermo Fisher Scien-tific). The cells were maintained at 37 °C in a humidified atmosphere of 95% air and 5% $CO_2$.

## Plasmids

For the full-length EGFP-Teneurin2 clone, partial Teneurin2 from KAZUSA cDNA (NCBI AB032953) and RIKEN cDNA (NCBI AK031198) were amplified by PCR and inserted into pEGFP-C1 or pmCherry-C1 (Takara Bio). The missing part was complemented by a custom gene synthesis (Eurofins Genomics). The protein translated from this plasmid is equivalent to the full-length *Homo sapiens* Teneurin-2 (NCBI NP_001382389), consisting of 2774 amino acids. Note that there are six mutations in our construction as follows: I418V, M431V, V590L, S659A, T720S, and L2384P. They are not located in the intracellular domain. EB1-TagRFP was generated by PCR amplification from KAZUSA cDNA (NCBI AB463888) and inserted into pTagRFP-N (Evrogen). For EB3-EGFP, the EB3 sequence was artificially synthesized (Eurofins Genomics) and inserted into EGFP-N1 (Takara Bio). GST-EB1 and GST-EB3 were amplified by PCR from EB1 and EB3 sequence, respectively, and inserted into the pGEX-6P3 vector (Cytiva). LifeAct-TagRFP was created by inserting a sequence encoding a 17-amino acid peptide derived from yeast Abp140 into the NheI-BamHI site of pTagRFP-N (*Riedl et al., 2008*). pHluorin-GABA$_A$Rγ2 was a gift from Tija Jacob & Stephen Moss (Addgene plasmid # 49170; http://n2t.net/addgene:49170; RRID:Addgene_49170). The shRNA target sequence was designed for protein knockdown using the BLOCK-iT RNAi Designer tool (Thermo Fisher Scientific). A cassette containing the pre-shRNA sequence was inserted into pBAsi-mU6 (Takara Bio). The target sequences of each shRNA are as follows: Negative control, GCCTAAGGTTAAGTCGCC; Teneurin2 #1, GCCAGGTTTGATTATACC. For volume marker, the SV40 promoter and tagRFP sequences were amplified and inserted into the pBAsi-mU6 vector.

## Transfection

COS-7 cells were transfected with the plasmid using Lipofectamine 2000 (Thermo Fisher Scientific) following manufacturer's protocol. Cultured neurons were transfected with the High-Efficiency Ca$^{2+}$ Phosphate Transfection Kit (Takara Bio), as per the manufacturer's protocol. Briefly, the culture medium was replaced with fresh MEM containing pyruvate, glucose, and GlutaMAX. Next, a mixture of 2 µg of plasmid, 3.1 µl of 2 M CaCl$_2$, and 25 µl of Hanks equilibrium salt solution was prepared and incubated at room temperature for 15 min. The DNA/Ca$^{2+}$ phosphate suspension was then added to the culture medium and incubated in a 5% CO$_2$ incubator at 37 °C for 40 min to 1 hr. After the incubation, the DNA/Ca$^{2+}$ phosphate precipitates were dissolved for 15 min with pre-equilibrated medium in a 10% CO$_2$ incubator before being replaced with the original medium. The minimum detectable fluorescence intensity was selected for analysis, except for the co-localization assay with EB1 (*Figure 6C*).

## Antibodies

The TEN2-specific affinity-purified rabbit polyclonal antibody was generated against a synthetic peptide sequence CSNTSHQIMDTNPDE (Eurofins Genomics) located at the amino acids 203–216 in the intracellular domain with an additional cysteine at the N-terminal. To purify the antibody, SulfoLink Coupling Resin (Thermo Fisher Scientific) with a synthetic peptide sequence CQMPLLDSNTSHQIMD TNPDEEFSPNS (GenScript), which corresponds to amino acids 196–222 in the intracellular domain, was used after crude purification of serum with ammonium sulfate. For other antibodies, commercially available antibodies were used, as indicated in Key Resources Table. When immunized animals overlapped and could not be labeled with secondary antibodies simultaneously, labeling was performed by non-covalent antibody labeling kits FlexAble (Proteintech) or Zenon (Thermo Fisher Scientific) following the manufacturer's protocol prior to immunostaining.

## Immunostaining

Cells were first washed with PBS at 37 °C and then fixed with 4% paraformaldehyde for 20 min. Next, they were permeabilized with 0.1% Triton X-100 for 3 min and blocked with 5 or 10% bovine serum albumin (BSA, Merck) in PBS for 20 min. Primary antibodies were diluted in Can Get Signal Solution (Toyobo) and incubated with the cells for 1 hr at room temperature, followed by incubation with secondary antibodies for 1 hr at room temperature. Actin was visualized with Alexa Fluor 555 Phalloidin (Thermo Fisher Scientific, 1:200) at the same time as probing with the secondary antibody. For detection of antigens on the membrane surface of live cells, anti-HA antibody was diluted at 1:250 in a culture medium and reacted for 30 min in a cell culture incubator. For detection of total/ surface TEN2, an anti-HA antibody pre-labeled with FlexAble was used. Additionally, PFA fixation,

detergent permeabilization, blocking treatment, and other primary antibody reactions were carried out as usual, followed by incubation with the secondary antibody. Immunostaining images were mainly acquired using a confocal laser scanning microscope (LSM 880; ZEISS) equipped with a 63×/1.4 Plan Apochromat oil immersion objective or an IX71 inverted microscope (Olympus) equipped with Delta-Vision (Cytiva), a CoolSNAP HQ2 CCD camera (Teledyne Photometrics), and a 40×/1.35 UApo/340 oil immersion objective.

## dSTROM

dSTORM was performed using a Nikon Eclipse Ti inverted microscope equipped with sapphire lasers of 488 nm and 561 nm (Coherent), a 647 nm fiber laser (MPB Communications), and a Plan Apo TIRF 100×Oil Immersion objective. The perfect focus system (PFS) was used to maintain focus during recording. The observations were conducted in a buffer containing 50 mM Tris-HCl (pH 8.0), 10 mM NaCl, 0.1 M MEA, 0.7 mg/ml glucose oxidase, 10% glucose, and 0.034 mg/ml catalase. The observations were taken continuously at 59 Hz for Alexa Fluor 647 and CF568 in that order, and 25,000 images were recorded for each dye. All recordings were drift-corrected, and STORM images were constructed using Nikon's accompanying analysis software, NIS.

## Proximity ligation assay

The proximity ligation assay was performed using DuoLink (Merck) and following the protocol provided by the supplier. Primary antibodies were applied according to the immunostaining protocol described above. PLA Probe Anti-Mouse PLUS (Merck) and PLA Probe Anti-Rabbit MINUS (Merck) were diluted 1:10 and incubated at 37 °C for 1 hr. After three washes, oligonucleotides labeled with the two secondary antibodies were ligated with ligase, and DNA was amplified by rolling circle amplification. The newly synthesized DNA was then labeled with Duolink in situ detection reagent green (Merck). After three more washes and a 5 min post-fixation step, neurons were incubated with a secondary antibody against MAP2 for 1 hr. Observations were made after three additional washes.

## Live imaging and image analysis

Images were analyzed using FIJI/ImageJ. Live observations were recorded using a confocal laser scanning microscope (LSM 880; ZEISS) equipped with a 63×/1.4 Plan Apochromat oil immersion objective with a resolution of 512×150 and a pixel time of 16.48 µs. The frame intervals were 2.97 s for EB3-EGFP and 15 s for pHluorin-GABA$_A$Rγ2. During observation, the medium was replaced with Leibovitz L-15 medium (FUJIFILM Wako) and maintained at 37 °C in a homemade temperature-controlled chamber. Kymographs were created using the KymoResliceWide plugin (Eugene Katrukha) or Kymograph Clear 2.0 a (*Mangeol et al., 2016*). Since dendrites have a certain thickness, the analysis was limited to 5 pixels (6.6 µm) from the periphery. Out-of-focus areas in the field of view were manually excluded. The detection of comets was automatic in Kymograph Clear 2.0 a and manual in KymoResliceWide. For analysis of GABA$_A$Rγ2, the time-dependent changes in brightness in each ROI were measured by designating the region of interest (ROI) using ImageJ, which had a constant level of pre-bleached pHluorin channel. The overlay of live imaging and fixed image was automatically adjusted for coordinate alignment using ImageJ to maximize the correlation in the overlay channel.

To correct for differences in the number of gephyrin per neuron, we can calculate the pausing probability of EB comets ($p_{GEP}$) using the following formula:

$$p_{GEP} = 1 / (((1/r_{GEP} - 1)/(1 - l_{GEP})) * l_{GEP} + 1)$$

where $l_{GEP}$ is the ratio of gephyrin-positive length (gephyrin positive length / total observation length), and $r_{GEP}$ is the ratio of comets paused in gephyrin-positive regions (number of comets paused at gephyrin / number of total observed comets). Alternatively, we can express $r_{GEP}$ in terms of $p_{GEP}$ and $l_{GEP}$:

$$r_{GEP} = p_{GEP} * l_{GEP} / (p_{GEP} * l_{GEP} + (1 p_{GEP}) * (1 l_{GEP}))$$

## Quantification of gephyrin puncta

To visualize gephyrin, we used an anti-gephyrin antibody. To trace dendrites, we expressed TagRFP in knockdown experiments and used MAP2 immunostaining in other experiments. We binarized the TagRFP or MAP2 channels to enhance dendritic visualization and traced the dendrites using the NeuronJ plugin. We quantified the number of gephyrin puncta using the SynapCountJ plugin, which required two images. To identify gephyrin puncta, we duplicated the gephyrin channel and used the same threshold for both channels.

## Quantitative analysis of correlation coefficient

For the analysis of TEN2-EB1 colocalization, COS-7 cells were plated in a Lab-Tek II 8-well chamber coverglasses at $1–2×10^5$ / well. 0.5 µg of pEGFP-C1 vector, which inserted the necessary part of TEN2, and 1 µg of EB1-TagRFP, and 0.5 µL Lipofectamin 2000 were diluted in Opti-MEM (Thermo Fisher Scientific) and added to cells. After 18 hr, cells were fixed with PFA and cells expressing EB1 in MT pattern were recorded with LSM 880. To exclude regions with no signal and full of noise, EGFP channels were binarized to detect cell morphology, and these were used as regions of interest (ROI). Since the background was sufficiently reduced under these conditions, quantitative analysis was performed using the non-threshold Pearson's R value as the correlation coefficient between TEN2 and EB1. For the analysis of TEN2-MAP2 colocalization, neurons expressing EGFP-TEN2 were immunostained with MAP2. The signal intensities of EGFP and MAP2 perpendicular to the direction of dendrite elongation were quantified using ImageJ's Plot Profile.

## Cluster analysis

For the cluster analysis, the number of clusters was predetermined to be 3. All postsynapses within 100 µm of the cell body of the eight neurons were included, and postsynapses were detected using a threshold of 32768 with 16 bits of gephyrin fluorescence intensity based on the same criteria. All observed samples were used for statistics, and no samples were excluded. The Kolmogorov-Smirnov test was used to test for normal distribution prior to significance testing. Nonparametric methods were used to analyze samples that were not normally distributed. T-tests were based on the Welch method and did not assume equal variance. The experimenter was not blinded to the experimental conditions during data acquisition or quantification.

## Motif search and alignment

The columns containing the Uniprot IDs of proteins from the table of Excitatory and Inhibitory Synaptic Cleft Proteomes (*Loh et al., 2016*) were converted to comma-separated csv files and then converted to fasta format using NCBI E-utilities. Fuzzpro in the EMBOSS package were used for each motif search for the following conditions: pattern Sx[ILV]P, mismatch 0 for SxφP motif; pattern LRPPTP[ILV], mismatch 2 for LxxPTPφ motif. "x" represents any amino acid. This search was performed in a macOS terminal with Anaconda installed. The obtained sequences were searched with Uniprot and manually checked whether they were extracellular or cytosolic. The alignment of amino acid sequences was performed using Clustal Omega (EMBL-EBI) and MacVector (MacVector).

## Electrophysiology

Whole-cell patch-clamp recordings were performed with a MultiClamp 700B amplifier (Molecular Devices, USA) in voltage-clamp mode (a holding potential of –70 mV) at room temperature from primary cultured hippocampal neurons at 15–17 DIV using patch pipettes (1–3 MΩ) pulled from Corning #0010 glass (PG10150-4, World Precision Instruments) as described previously (*Ninomiya et al., 2022*). TagRFP-positive neurons were selected for the recordings using a spinning disk confocal unit (CSU-X1, Yokogawa Electric, Tokyo, Japan) attached to an upright microscope (BX51WI, Olympus, Tokyo, Japan) as described previously (*Hoshino et al., 2021*). The pipette solution contained (in mM) 125 Cs-gluconate, 12.5 Cs-MeSO₄, 10 HEPES, 5 Mg-ATP, 0.5 Na-GTP, 1 EGTA, 2 QX-314 and 5 phosphocreatine (pH 7.3). The liquid junction potential was not corrected. The extracellular solution contained (in mM) 125 NaCl, 2.5 KCl, 1.25 $NaH_2PO_4$, 26 $NaHCO_3$, 2 $CaCl_2$, 1 $MgCl_2$, and 10 glucose (pH 7.4) and bubbled continuously with a mixture of 95% $O_2$ and 5% $CO_2$ at room temperature. Spontaneous mIPSCs were recorded in the presence of 1 µM TTX, 10 µM NBQX, and 50 µM D-AP5 that block action potential firing and excitatory synaptic currents mediated by AMPA receptors and NMDA

receptors, respectively. In this experimental condition, all the recorded synaptic currents were considered as GABAergic, because addition of 40 µM bicuculline, which blocks $GABA_A$ receptors, completely abolished spontaneous synaptic currents (n=3 neurons). The recorded currents were filtered at 2 kHz and acquired at 5 kHz using pCLAMP10 or pCLAMP11 software (Molecular Devices). mIPSCs were detected and analyzed using Igor Pro 8 (Wavemetrics) equipped with NeuroMatic software (http://www.neuromatic.thinkrandom.com/) (*Rothman and Silver, 2018*) and custom-written Igor procedures by NH. The algorithm of mIPSC detection was a threshold-above-baseline detector implemented in NeuroMatic software, and continuous 300 s records were used for the event detection. The detection threshold was mostly more than three times the standard deviation of event-free baseline noise, and all the detected events were checked by eye.

## GST pull-down

Recombinant proteins were expressed in *Escherichia coli* BL21(DE3) (Agilent). The cells were cultured in Lennox LB medium (Merck) containing ampicillin at 37 °C and induced with 0.6 mM isopropyl-β-D-thiogalactopyranoside (IPTG) for 18 hr at 16 °C. The harvested cells were lysed by sonication (UD-201; TOMY DIGITAL BIOLOGY) in a purification buffer (50 mM Tris-HCl, 50 mM NaCl, pH 8.0) and the supernatant was collected after centrifugation. The supernatant was then purified using Glutathione 4B Sepharose beads (Cytiva). Brain lysate was obtained from the brains of 4-month-old male knock-in mice after cervical dislocation. Brain was homogenized with a Potter homogenizer in pull-down buffer (20 mM Hepes, 150 mM NaCl, 0.5% TritonX-100, pH 7.4) containing Complete Mini protease inhibitor (Merck). The obtained brain lysate was used as the input after centrifugation at 100×*g* for 10 min and 20,000×*g* for 20 min. The purified GST fusion protein and Sepharose beads were equilibrated in the pull-down buffer and then mixed with the input at 4 °C for 1 hr under agitation. The Sepharose beads were washed three times with pull-down buffer and then eluted with 2×sample buffer. Proteins were denatured by heating samples for 10 min at 95 °C.

## SDS-PAGE and Western blotting

SDS-PAGE was performed using homemade gels consisting of a 3% acrylamide stacking gel and a 7.5% or 10% acrylamide running gel. Electrophoresis was conducted under conditions of upper limitations with 30 mA and 250 V for 1 hr. For gel staining, the gel was stained overnight in staining solution (0.25% CBB-R, 10% acetic acid). For western blotting, semi-dry blotting was performed, and the proteins were transferred to a PVDF Immobilon-P membrane (Merck) under constant current conditions of 3.7 mA/cm$^2$ for 40 min. The membrane was dried completely, then reactivated with methanol and equilibrated with TBS-T (50 mM Tris-HCl, 150 mM NaCl, 0.5% Tween-20). Subsequently, the membrane was blocked with 5% skim milk (Morinaga Milk Industry) in TBS-T for 20 min. Primary and secondary antibodies were diluted in Can Get Signal Solution (Toyobo). Membrane was incubated with each diluted solution for 1 hr at room temperature. After washing with TBS-T, chemiluminescence was performed using Immunostar Zeta (FUJIFILM Wako). Detection and quantification were performed using a LAS4010 imager (Cytiva).

## Acknowledgements

We thank Drs. Tohru Murakami and Yuki Tajika for technical advice. We thank Yoshihiro Morimura, Toshie Kakinuma, and Sachiko Sato for technical assistance and care of the mice. This work was supported by a Grant-in-Aid for Scientific Research (C) (18K06499, 22K06805, H.I) and Young Scientist (20K16104, SI) from the Ministry of Education, Culture, Sports, Science and Technology of Japan, and Takeda Science Foundation. This work used equipment shared in the MEXT Project for promoting public utilization of advanced research infrastructure (JPMXS0420600119, JPMXS0420600120, JPMXS0420600121).

# Additional information

## Funding

| Funder | Grant reference number | Author |
| --- | --- | --- |
| Japan Society for the Promotion of Science | Grant-in-Aid for Scientific Research (C) 18K06499 | Hirohide Iwasaki |
| Japan Society for the Promotion of Science | Grant-in-Aid for Scientific Research (C) 22K06805 | Hirohide Iwasaki |
| Takeda Foundation | Medical Research Grants | Hirohide Iwasaki |
| Japan Society for the Promotion of Science | Grant-in-Aid for Young Scientist 20K16104 | Sotaro Ichinose |

The funders had no role in study design, data collection and interpretation, or the decision to submit the work for publication.

## Author contributions

Sotaro Ichinose, Conceptualization, Data curation, Software, Formal analysis, Funding acquisition, Validation, Investigation, Visualization, Methodology, Writing - original draft, Project administration, Writing - review and editing; Yoshihiro Susuki, Validation, Investigation, Visualization; Nobutake Hosoi, Investigation, Visualization, Writing - original draft, Writing - review and editing; Ryosuke Kaneko, Resources; Mizuho Ebihara, Formal analysis, Validation, Investigation; Hirokazu Hirai, Investigation, Writing - review and editing; Hirohide Iwasaki, Conceptualization, Supervision, Funding acquisition, Project administration, Writing - review and editing

## Author ORCIDs

Sotaro Ichinose ⓘ http://orcid.org/0000-0002-1470-2957
Hirohide Iwasaki ⓘ http://orcid.org/0000-0002-7432-5938

## Ethics

The experiments in this study have passed a rigorous ethical review and have been approved by Gunma University for animal experiments (approval number: 20-061) .

## Decision letter and Author response

Decision letter https://doi.org/10.7554/eLife.83276.sa1
Author response https://doi.org/10.7554/eLife.83276.sa2

# Additional files

## Supplementary files

• Supplementary file 1. The results of motif search. In order to narrow down the candidates for MT recruiter, a motif search was conducted to investigate the potential binding with EB. Based on previous proteomics studies (*Loh et al., 2016*), proteins containing the motifs SxφP and LxxPTPφ were searched. The cellular localization (extracellular or intracellular) of each motif was manually checked.

• MDAR checklist

## Data availability

Excel files are attached as source files.

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
