## [Editor Report]

Ichinose and coauthors investigate the mechanisms that contribute to building inhibitory synapses through differential protein release from microtubules. In their valuable study, they find that teneurin-2 plays a role in this process in cultured hippocampal neurons via EB1 using a variety of genetic and imaging methods. The methods, data, and analysis are solid, and the manuscript will be of interest to neuroscientists and cell biologists interested in intracellular trafficking and synapse maturation.

---

## [Decision Letter]

**Decision letter after peer review:**

Thank you for submitting your article "Teneurin-2 at the Synapse Construction Site is a Signpost for Cargo Unloading from Motor Proteins" for consideration by *eLife*. Your article has been reviewed by 3 peer reviewers, and the evaluation has been overseen by a Reviewing Editor and Anna Akhmanova as the Senior Editor. The reviewers have opted to remain anonymous.

Essential revisions:

After extensive discussion, it is clear that the authors must perform a number of experiments to strengthen their paper and make substantial textual changes to present their findings in a consistent and valid way. We highly recommend that you address each point of all three reviews and perform the necessary experiments, if feasible, and edit the text accordingly. However, the major experiments agreed upon by all three reviewers are below.

1. All three reviewers agreed that some of the major overstatements by the authors are also contradictory and these need to be addressed. One major question is how TEN2 is recruiting microtubules to the membrane. Additional experiments are necessary to address this if the authors do indeed believe it is through EB1. It was suggested by the reviewers that since the authors have access to the HA-TEN2 neurons, many of the COS7 cell experiments should be performed in this more relevant neuronal model. In addition, the authors should perform pull-down experiments to show an interaction between TEN-2 and EB1. Ideally, the pull-down assays should be performed from the brain lysates of the HA-TEN2 mouse model.

2. Reviewer 1 suggested that cargo transport assays, like GABAr trafficking as shown in Twelvetrees et al., 2010 need to be performed to arrive at any conclusion about cargo unloading at TEN-2 sites.

3. As a proper negative control, TEN2N-L2mut, not TEN2TM, should be used for the experiments presented in Figures 3 and 6.

4. All three reviewers agreed that the authors should take the advantage of the HA-TEN2 mouse model. First, the authors must validate the model by at least performing colocalization studies between HA and TEN2 antibodies via immunocytochemistry. It should also be straightforward for the authors to discern the subcellular localization of TEN2 (intracellular versus plasma membrane) at each stage/cluster by combining surface and intracellular staining.

5. All three reviewers agreed that the authors should perform electrophysiology to determine the actual role of TEN2 in a functional context. Please refer to Reviewer 2's comment. If you are not technically able to perform this experiment, please include a statement about this limitation in the paper.

6. All three reviewers discussed the importance of controlling for the expression levels of all versions of TEN2 because some of the results could be misconstrued if the expression levels of the different constructs are different. Please provide the necessary control western blots.

7. Please refer to the reviewers' comments on textual changes that will help bolster the findings and strengthen the story. In addition, please display the best images possible for the manuscript. There were concerns that the readers would not be able to see the result stated by the authors in some of the images provided.

*Reviewer #1 (Recommendations for the authors):*

Additional experiments:

– While there may be support to the idea that end binding proteins facilitate cargo release in literature, several experiments need to be done by the authors before they can make the same conclusion for TEN-2 and its potential interaction with EB1 in neurons. Firstly, the current evidence for a direct link between EB1 plus ends and TEN-2 in neurons is quite weak and could be strengthened by assaying (a) EB1 dynamics in the shTEN2 and TEN2N-L dominant negative conditions and (b) pull-down experiments to show an interaction between TEN-2 and EB1. Second, cargo transport assays, like GABAr trafficking as shown in Twelvetrees et al., 2010 for example, need to be performed to arrive at any conclusion about cargo unloading at TEN-2 sites.

Experimental methods:

– It is unclear how actin is labeled in Figure 1 experiments.

– The motif search process is well explained in the text of the Results section, it seems unnecessary to devote two separate figures (Figure 1-supplement 2B and figure 1D) to it.

– In figure 3G, it is unclear which region was used to analyze the number of comets. Was it the entire cell or a predefined ROI?

– Figure 6D portrays a ratio of neurons, but it is unclear from the text or legends as to what the authors mean by that.

Writing:

– MT and MTs are used interchangeably throughout the text. For example, line 22 in the abstract should read "MT recruitment" instead of "MTs recruitment".

– The introduction and the Results section repeat the same paragraph: lines 38-42 and lines 93-97 are nearly identical.

– In the introduction, how the authors arrive at their hypothesis (line 73) is unclear from the writing since there is no reference to the relationship between teneurins and the cytoskeleton from previous work (eg., Mosca et al., 2012).

– Is there a previous classification of the semi-periphery region of inhibitory post synapses?

– Reword line 245 to " observed precise localization using a form of super-resolution microscopy (SRM)".

– Line 268 "and gephyrin were not always perfectly colocalized".

– Line 310 is not supported by evidence.

– The Discussion section has portions that are not relevant to the results presented and can be cut short.

*Reviewer #2 (Recommendations for the authors):*

The present paper clearly has potential, but it appears premature in its current form. Although it has documented several interesting findings, there are some loose ends listed below.

1. The results of knockdown and dominant-negative (DN) approaches suggest that TEN2 is important for inhibitory synapse formation. However, it is necessary to identify which stage(s)/cluster(s) of the inhibitory synapse complex is regulated by TEN2. This should be addressed by the cluster analysis against TEN2 KD and TEN2-DN overexpressed neurons. In addition, it is important to address how extracellular and intracellular TEN2 domains regulate cluster 3 synapse formation. The authors often refer that TEN2 "promotes" receptor accumulation (rows 21, 78, 306, 411) which cannot be addressed just by loss-of-function approaches. The authors should test whether full-length TEN2 overexpression has a gain-of-function effect in cluster 3 synapse formation.

TEN2N-L2mut, not TEN2TM, should be chosen as a proper negative control. It is recommended to test TEN2N-L2mut in at least a few critical experiments in Figures 3 and 6.

2. The rationale for testing the HA-TEN2 KI mouse model is weak and I don't think that the authors take the advantage of this mouse model. Performing MT-trapping assay as in Figure 3 to HA-TEN2 KI primary neurons should be able to address the TEN2 function in neurons. The authors hypothesize that TEN2 protein is critical for the transition from cluster 1 to 2. However, it is not clear whether cytosolic or membrane-targeted TEN2 plays a role in this step. It'll be very interesting to identify the subcellular localization of TEN2 (intracellular versus plasma membrane) at each stage/cluster by combining surface and intracellular staining. I do not expect all of these additional experiments to be performed but some further information would strengthen the manuscript.

It is important to perform the proper validation of the HA-TEN2 mouse model. It is essential to present the colocalization of HA and ICD antibody signals in immunocytochemistry and immunoblotting.

3. It is interesting that shTenm2 has a moderate effect on GABAAR expression. In which cluster are the receptors located? Are they targeted to the plasma membrane or stayed in cluster 1? Performing electrophysiology recording in addition to the cluster analysis should give a clear-cut result to understand the roles of TEN2 in functional inhibitory synapse formation.

4. Although gephyrin puncta are found in all clusters (Figure 1C), it is not clear how gephyrin is transferred in the models presented in (Figures 1-supplement 2 and 7). The model should be based on experimental results.

5. Overstatement is scattered. For example, "rows 25-27: Our study revealed that cargo release from kinesins through TEN2-MTs interactions supports the continuity from partner choice to synaptogenesis, which is a critical step in synaptic maturation." This study fully focuses on TEN2 function in inhibitory synapse development, and kinesin and cargo dynamics are not studied. Re-editing the manuscript is recommended.

6. Figure 2F should include the same plot against TEN2TM for comparison.

7. Rows 190-191: The resolution of Figure 3B is too low to get the authors' conclusion that the EB1 binding motif was correctly located in the cytoplasm.

8. Sample images that present the localization of TEN2 in each cluster, like Figure 1C, should be presented.

*Reviewer #3 (Recommendations for the authors):*

Introduction

1) The hypothesis in in the introduction (line 73) seems out of place. This may be better posed as a question and the sentence should be simplified for clarity.

2) The introduction leads the reader to expect that the authors will have solved some aspect of E vs I synapse formation, when in fact this paper deals only with I synapses. Reviewer suggests focusing the introduction on the question of either adhesion molecules in synapses OR how kinesins release cargo as posed in the abstract. Less is more here and throughout the paper.

3) Line- 38- The authors should note that not all excitatory synapses form on spines.

4) Line 39- The authors should draw clearer distinctions between the cytoskeletons of excitatory versus inhibitory synapses. As written, no clear difference is presented.

5) Lines 46-53. The authors should consider including a schematic that shows inhibitory synapse construction in Figure 1A.

Line 54- The first sentence of this paragraph is a bit confusing and unnecessary

6) Line 58-60, revise to read: "Teneurin-2 (TEN2) is one of the few molecules that has been suggested to function in a continuity from synaptic specificity to synaptogenesis".

7) Line 63, SS, should be changed to SS-.

8) Lines 118- 127. This section is confusing and should be simplified. Reviewer suggests condensing into 1-2 sentences that introduce the next paragraph on MT recruiters.

9) Lines 92-99- The results should not repeat information already given in the introduction.

10) Lines 164-172. Simplify to 1-2 sentences.

11) Figure 2E TEN2TM- authors may consider replacing with a better-resolved image as image quality seems to differ from that of TEN2N-L.

12) Line 320-328. The authors should clarify that TEN2 is tagged with EGFP and that's why EGFP was measured. The authors should also better justify this experiment to provide evidence that differential extraction with saponin indeed indicate differential association with the cytoskeleton.

---

## [Author Response]

Essential revisions:After extensive discussion, it is clear that the authors must perform a number of experiments to strengthen their paper and make substantial textual changes to present their findings in a consistent and valid way. We highly recommend that you address each point of all three reviews and perform the necessary experiments, if feasible, and edit the text accordingly. However, the major experiments agreed upon by all three reviewers are below.

First, we would like to thank Dr. Anna Akhmanova, Senior Editor, Dr. Kassandra Ori-McKenney, Reviewing Editor, and the anonymous reviewers for taking the time out of their busy schedules to review our manuscript. Your expertise and insightful feedback were invaluable in shaping the submitted version of the manuscript. Thank you for your diligent efforts and constructive comments.

1. All three reviewers agreed that some of the major overstatements by the authors are also contradictory and these need to be addressed. One major question is how TEN2 is recruiting microtubules to the membrane. Additional experiments are necessary to address this if the authors do indeed believe it is through EB1. It was suggested by the reviewers that since the authors have access to the HA-TEN2 neurons, many of the COS7 cell experiments should be performed in this more relevant neuronal model. In addition, the authors should perform pull-down experiments to show an interaction between TEN-2 and EB1. Ideally, the pull-down assays should be performed from the brain lysates of the HA-TEN2 mouse model.

As suggested, live imaging of EBs was performed in the neuron (Figures 1C-H and 7A-E). In addition, pull-down assay was performed using brain lysate of HA knock-in mice to detect the interaction between EB1/3 and TEN2 (Figure 6).

2. Reviewer 1 suggested that cargo transport assays, like GABAr trafficking as shown in Twelvetrees et al., 2010 need to be performed to arrive at any conclusion about cargo unloading at TEN-2 sites.

We first performed live imaging of EGFP-GABAARγ2 in the neuron (data not shown). We noticed the presence of several types of vesicles, which may contain newly synthesized receptors as well as endocytosed vesicles. We felt that we needed to observe only the newly synthesized receptors. Instead, we performed a FRAP assay with pHluorin-GABAARγ2 to detect "exocytotic platforms." As a result, we were able to observe exocytosis of GABAARγ2 at TEN2-positive positions (Figure 4G-J).

3. As a proper negative control, TEN2N-L2mut, not TEN2TM, should be used for the experiments presented in Figures 3 and 6.

The previous version of Figure 6 was retested using TEN2N-L2mut together (Figures 7F-H). We have removed the previous version of Figure 3 in order to appropriately incorporate the improved EB1 capture by immobilized TEN2 in neurons as well as COS-7 cells, as mentioned in point #1 above. However, at present, culturing neurons on cover glasses coated with HA-antibody is not feasible due to viability issues. Therefore, this issue remains for future investigation, and alternative methods, such as measuring kinetic force, need to be employed for validation. This matter is also discussed in the Discussion section (line 415-422).

4. All three reviewers agreed that the authors should take the advantage of the HA-TEN2 mouse model. First, the authors must validate the model by at least performing colocalization studies between HA and TEN2 antibodies via immunocytochemistry. It should also be straightforward for the authors to discern the subcellular localization of TEN2 (intracellular versus plasma membrane) at each stage/cluster by combining surface and intracellular staining.

Thank you for valuable suggestion. We have performed the co-localization analysis of HA and TEN2ICD, as suggested, and the results are presented in Figure 3 —figure supplement 1E-G. The co-localization is observed to be moderate, which could be attributed to the fact that the anti-TEN2ICD antibody used in the study is a structure-recognition antibody, thereby selectively detecting a specific population of TEN2. Additionally, we have also included the extra/intracellular localization patterns for each developmental stage in Figures 3A-D.

5. All three reviewers agreed that the authors should perform electrophysiology to determine the actual role of TEN2 in a functional context. Please refer to Reviewer 2's comment. If you are not technically able to perform this experiment, please include a statement about this limitation in the paper.

As suggested, we conducted electrophysiological experiments (Figure 5). Our analysis of miniature inhibitory postsynaptic currents (mIPSCs) revealed a significant prolongation in the inter-event interval of mIPSCs in TEN2 knockdown neurons. However, we observed no significant difference in the amplitude of mIPSCs between the control and knockdown neurons. These findings suggest that TEN2 knockdown leads to a decrease in the frequency of mIPSCs without affecting the individual synaptic strength.

6. All three reviewers discussed the importance of controlling for the expression levels of all versions of TEN2 because some of the results could be misconstrued if the expression levels of the different constructs are different. Please provide the necessary control western blots.

In the previous version, it was suggested that the interpretation of certain data could be confusing due to differences in the expression levels of EGFP-TEN2. We have thoroughly considered this matter, but due to insufficient quantification and reproducibility, we have decided to withdraw these data. However, we have conducted new experiments using live imaging, taking sufficient care for quantification (Figures 1D-H, 4G-J, 7A-E). Therefore, we believe that we can present reliable data.

7. Please refer to the reviewers' comments on textual changes that will help bolster the findings and strengthen the story. In addition, please display the best images possible for the manuscript. There were concerns that the readers would not be able to see the result stated by the authors in some of the images provided.

We have extensively revised both the text and figures, focusing on narrowing down the central points of discussion. We believe that this version adequately meets the requirements and expectations.

Reviewer #1 (Recommendations for the authors):Additional experiments:– While there may be support to the idea that end binding proteins facilitate cargo release in literature, several experiments need to be done by the authors before they can make the same conclusion for TEN-2 and its potential interaction with EB1 in neurons. Firstly, the current evidence for a direct link between EB1 plus ends and TEN-2 in neurons is quite weak and could be strengthened by assaying (a) EB1 dynamics in the shTEN2 and TEN2N-L dominant negative conditions and (b) pull-down experiments to show an interaction between TEN-2 and EB1. Second, cargo transport assays, like GABAr trafficking as shown in Twelvetrees et al., 2010 for example, need to be performed to arrive at any conclusion about cargo unloading at TEN-2 sites.

Thank you once again for providing constructive suggestions. We have strengthened the evidence by conducting additional experiments, as Reviewer #1 proposed, including observing EB dynamics under TEN2N-L dominant-negative conditions and performing pull-down experiments to demonstrate the interaction between TEN-2 and EB1. However, we determined that it is not appropriate to perform cargo transport assays, such as GABAAR transport in Twelvetrees et al., 2010, because we cannot distinguish between newly synthesized proteins and endocytosed ones. In our study, it is crucial to differentiate between these two vesicles, as mixed quantification could lead the misinterpretation. Therefore, direct observation of cargo unloading from kinesin at TEN-2 sites was not possible. Instead, we evaluated exocytosis at TEN-2 positions using FRAP assays with pHluorin-GABAAR (Figure 4G-J). The results showed a significant increase in GABAAR exocytosis at TEN-2-positive positions. Although we could not provide direct evidence of "cargo unloading," we should be able to claim that "TEN2 provides a platform for exocytosis." Therefore, we are reviewing the expression of our claims, including the title.

Regarding EB1 dynamics in shRNA experiments (we used EB3 in our experimental system due to poor signal/noise ratio for EB1 comet/diffusion in the dendrite), it was not feasible due to the following temporal inconsistencies: It takes 2-3 days for TEN2 knockdown after transfection, but by that time, the expression levels of EB3 have increased, leading to its localization on the MT lattice. Shifting the timing of shRNA and EB transfection (i.e., two-times transfection) decreased cell viability, so it was not possible to implement.

Experimental methods:– It is unclear how actin is labeled in Figure 1 experiments.

Thank you for your feedback. In cluster analysis, we used phalloidin. However, in other experiments, we also used LifeAct for visualization. Therefore, we have explicitly mentioned both methods.

– The motif search process is well explained in the text of the Results section, it seems unnecessary to devote two separate figures (Figure 1-supplement 2B and figure 1D) to it.

In the current version, we have included Figure 2A (previously shown in Figure 1D) only.

– In figure 3G, it is unclear which region was used to analyze the number of comets. Was it the entire cell or a predefined ROI?

This analysis has been removed in the current version, but it was performed by setting the region of interest (ROI) to the entire cell.

– Figure 6D portrays a ratio of neurons, but it is unclear from the text or legends as to what the authors mean by that.

In neurons transfected with TEN2N-L, we observed that microtubules in dendrites were biased towards the periphery rather than the center. However, due to potential ambiguity in the presentation and description, we have included this information in the supplement (Figure S7).

Writing:– MT and MTs are used interchangeably throughout the text. For example, line 22 in the abstract should read "MT recruitment" instead of "MTs recruitment".

We have unified the usage of "MT" as an adjective and "MTs" as a noun.

– The introduction and the Results section repeat the same paragraph: lines 38-42 and lines 93-97 are nearly identical.

Thank you for the feedback. We have taken care to avoid duplication and ensure that the content is not redundant.

– In the introduction, how the authors arrive at their hypothesis (line 73) is unclear from the writing since there is no reference to the relationship between teneurins and the cytoskeleton from previous work (eg., Mosca et al., 2012).

We have provided a specific explanation with the citation (lines 81-87).

– Is there a previous classification of the semi-periphery region of inhibitory post synapses?

There have been studies suggesting the presence of multiple domains at the nanoscale within synapses, indicating that synapses are not uniform. While this phenomenon has been extensively reported in excitatory synapses, it is still an emerging topic in inhibitory synapses. We have cited a representative study (Yang et al., 2021) from this recent progression.

– Reword line 245 to " observed precise localization using a form of super-resolution microscopy (SRM)".

Thank you for the suggestion. The English text has been proofread as indicated in lines 206-208.

– Line 268 "and gephyrin were not always perfectly colocalized".

The English text has been proofread as indicated in lines 238-239.

– Line 310 is not supported by evidence.

As Reviewer #1 pointed out, the enhanced co-localization observed in COS-7 cells is solely based on the overexpression of EB and remains speculative in an endogenous context. Therefore, this sentence has been removed from the Results section.

– The Discussion section has portions that are not relevant to the results presented and can be cut short.

As you correctly pointed out, we have limited the content of the discussion to only what has been demonstrated through experiments. Please review the new version.

Reviewer #2 (Recommendations for the authors):The present paper clearly has potential, but it appears premature in its current form. Although it has documented several interesting findings, there are some loose ends listed below.1. The results of knockdown and dominant-negative (DN) approaches suggest that TEN2 is important for inhibitory synapse formation. However, it is necessary to identify which stage(s)/cluster(s) of the inhibitory synapse complex is regulated by TEN2. This should be addressed by the cluster analysis against TEN2 KD and TEN2-DN overexpressed neurons. In addition, it is important to address how extracellular and intracellular TEN2 domains regulate cluster 3 synapse formation. The authors often refer that TEN2 "promotes" receptor accumulation (rows 21, 78, 306, 411) which cannot be addressed just by loss-of-function approaches. The authors should test whether full-length TEN2 overexpression has a gain-of-function effect in cluster 3 synapse formation.TEN2N-L2mut, not TEN2TM, should be chosen as a proper negative control. It is recommended to test TEN2N-L2mut in at least a few critical experiments in Figures 3 and 6.

Thank you very much for your suggestions. We have incorporated cluster analysis to examine the localization tendencies of each synaptic organizer. However, recognizing that cytoskeletal molecules are dynamic and cannot be assessed solely in a fixed state, we conducted a new live imaging experiment to assess the impact of dominant-negative (Figure 7A-E). In knockdown experiments, there was a discrepancy in the time window because knockdown requires 3 days post-transfection, whereas the optimal period for live imaging is 1 day post-transfection. Furthermore, utilizing endogenous proteins is considered ideal. Overcoming these limitations and developing techniques to track the developmental stages of neurons remain challenges within the scientific community. We extensively discussed these matters in the Discussion section (lines 408-413, 425-461).

Regarding gain-of-function experiments, it seemed feasible to transfer the experiments conducted in non-neuronal cells (previous version of Figure 3 and Sando et al., 2019) to neurons. However, the immobilized TEN2 experiment we conducted in COS-7 cells could not be carried out in neurons due to the significantly reduced cell viability when culturing neurons on antibody-coated cover glasses. Induction of synaptic formation experiments, which are commonly performed using non-neuronal cells, showed that overexpression of TEN2 in neurons led to its retention in the ER, resulting in a different localization than endogenous TEN2. Therefore, determining how to achieve gain-of-function that reflects endogenous functionality remains a challenge.

We conducted additional experiments using TEN2N-L2mut as a negative control.

2. The rationale for testing the HA-TEN2 KI mouse model is weak and I don't think that the authors take the advantage of this mouse model. Performing MT-trapping assay as in Figure 3 to HA-TEN2 KI primary neurons should be able to address the TEN2 function in neurons. The authors hypothesize that TEN2 protein is critical for the transition from cluster 1 to 2. However, it is not clear whether cytosolic or membrane-targeted TEN2 plays a role in this step. It'll be very interesting to identify the subcellular localization of TEN2 (intracellular versus plasma membrane) at each stage/cluster by combining surface and intracellular staining. I do not expect all of these additional experiments to be performed but some further information would strengthen the manuscript.It is important to perform the proper validation of the HA-TEN2 mouse model. It is essential to present the colocalization of HA and ICD antibody signals in immunocytochemistry and immunoblotting.

We were unable to perform the MT-trapping assay due to the low viability of neurons cultured on antibody-coated coverslips. Therefore, we have postponed the claim regarding the enhancement of microtubule recruitment through the binding of TEN2 with its extracellular binding partner.

We have demonstrated the surface expression of TEN2 in regions excluding the cell body and proximal dendrites in Figures 3A-D and S3I. Additionally, we have shown that TEN2 provides a platform for GABAAR exocytosis in Figures 4G-J. We assumed in the previous version that the expression of TEN2 on the cell surface is required for the transition from Cluster 1 to Cluster 2, and we believe that this aspect has been demonstrated through the additional experiments. However, as mentioned in point 1 above, we are unable to show whether there is an actual change in the clusters to which TEN2 belongs before and after exocytosis because of different time windows required for live imaging.

The validation of antibodies was performed through immunocytochemistry (Figure S3E-G), where moderate co-localization was observed. Assuming there is no non-specific detection, it can be said that the antibody specifically recognizes a particular population of TEN2. Unfortunately, this TEN2ICD antibody did not detects TEN2 in Western blotting. These results suggest that the antibody may be a conformation-specific antibody. This is also mentioned in the Discussion section (lines 365-381).

3. It is interesting that shTenm2 has a moderate effect on GABAAR expression. In which cluster are the receptors located? Are they targeted to the plasma membrane or stayed in cluster 1? Performing electrophysiology recording in addition to the cluster analysis should give a clear-cut result to understand the roles of TEN2 in functional inhibitory synapse formation.

We sincerely appreciate the valuable feedback and the interest in cluster analysis. However, as mentioned above, we did not employ cluster analysis as a crucial tool due to the dynamics of cytoskeletal molecules. Instead, we opted for an alternative approach using FRAP assays with pHluorin-GABAARγ2, which led us to interpret that TEN2 acts as a platform for exocytosis (Figures 4G-J). Notably, we observed fluorescence recovery even in TEN2-negative puncta, suggesting the presence of functionally similar molecules. Additionally, our electrophysiological analysis revealed a decrease in the frequency of mIPSCs without affecting their amplitude (Figure 5). This implies the existence of intact synapses that are not dependent on TEN2. Immunostaining of knockdown cells also showed no significant difference in the quantity of α1. Therefore, we propose that the impact of TEN2 on exocytosis is specific to certain subunit compositions rather than a general exocytosis. It is this specificity that explains why TEN2 knockdown only results in a moderate loss rather than complete absence of inhibitory synapses.

4. Although gephyrin puncta are found in all clusters (Figure 1C), it is not clear how gephyrin is transferred in the models presented in Figures 1-supplement 2 and 7. The model should be based on experimental results.

Thank you for your suggestion. Based on Maas et al., 2009, we have added gephyrin to the transported receptors.

5. Overstatement is scattered. For example, "rows 25-27: Our study revealed that cargo release from kinesins through TEN2-MTs interactions supports the continuity from partner choice to synaptogenesis, which is a critical step in synaptic maturation." This study fully focuses on TEN2 function in inhibitory synapse development, and kinesin and cargo dynamics are not studied. Re-editing the manuscript is recommended.

Based on the results of additional experiments, we have revised the overstatement regarding several claims that could not be confirmed. We have narrowed down the claim to focus only on TEN2 providing an exocytosis platform.

6. Figure 2F should include the same plot against TEN2TM for comparison.

Because of the dynamics of EB, it is difficult to evaluate co-localization with EBs in fixed cells. Therefore, considering the limited quantifiability and reproducibility, we have made the decision to remove this experiment.

7. Rows 190-191: The resolution of Figure 3B is too low to get the authors' conclusion that the EB1 binding motif was correctly located in the cytoplasm.

This experiment was also excluded from the current publication as it could not be replicated in neurons. We appreciate the suggestion and will consider it for future publications.

8. Sample images that present the localization of TEN2 in each cluster, like Figure 1C, should be presented.

To ensure immediate recognition by readers, we have replaced the extensive images with sample images (Figure 2B in the current version).

Reviewer #3 (Recommendations for the authors):Introduction1) The hypothesis in in the introduction (line 73) seems out of place. This may be better posed as a question and the sentence should be simplified for clarity.

We have resolved uncertainties and improved clarity in the logical flow of the document. We sincerely appreciate the valuable feedback we received from the reviewer #3. Please note, however, that some of the points raised by reviewer #3 have already been removed during the revision process.

2) The introduction leads the reader to expect that the authors will have solved some aspect of E vs I synapse formation, when in fact this paper deals only with I synapses. Reviewer suggests focusing the introduction on the question of either adhesion molecules in synapses OR how kinesins release cargo as posed in the abstract. Less is more here and throughout the paper.

We admit that there were some points that were excessive or not adequately proven. Therefore, we have removed the inadequately supported aspects from this publication and narrowed our focus solely to the role of TEN2 as a platform for exocytosis. We have made efforts to eliminate redundancy as much as possible.

3) Line- 38- The authors should note that not all excitatory synapses form on spines.

We have made revisions as follows: Excitatory postsynapses mainly form and mature on characteristic structures called spines, which are composed predominantly of actin (lines 40-41).

4) Line 39- The authors should draw clearer distinctions between the cytoskeletons of excitatory versus inhibitory synapses. As written, no clear difference is presented.

We have provided a more detailed explanation to clarify any unclear points.

5) Lines 46-53. The authors should consider including a schematic that shows inhibitory synapse construction in Figure 1A.

It can be found in Figure S2A of current version.

Line 54- The first sentence of this paragraph is a bit confusing and unnecessary.

Thank you for your suggestion. This content has been removed as it was not directly relevant to this publication, because we narrowed down the topics to be claimed.

6) Line 58-60, revise to read: "Teneurin-2 (TEN2) is one of the few molecules that has been suggested to function in a continuity from synaptic specificity to synaptogenesis".

This content also has been removed as it was not directly relevant to this publication, because we narrowed down the topics to be claimed.

7) Line 63, SS, should be changed to SS-.

Thank you for your suggestion. We have checked for any typos.

8) Lines 118- 127. This section is confusing and should be simplified. Reviewer suggests condensing into 1-2 sentences that introduce the next paragraph on MT recruiters.

Based on the live imaging results of EB, which showed a high abundance of EB pausing at inhibitory synapses, it suggests the presence of MT recruiters and has led to a logical progression of investigating whether synaptic organizers possess this function.

9) Lines 92-99- The results should not repeat information already given in the introduction.

It has been removed from the Results section.

10) Lines 164-172. Simplify to 1-2 sentences.

We have made revisions as follows: Endogenous EB1 is localized to the plus ends of MTs and observed as dynamic comets. However, this localization is lost upon cell fixation. Therefore, we overexpressed EB1 and localized it throughout MTs to detect protein-protein interactions (Skube et al., 2010).

11) Figure 2E TEN2TM- authors may consider replacing with a better-resolved image as image quality seems to differ from that of TEN2N-L.

Thank you for your suggestion. We replaced the images with those from a different cell.

12) Line 320-328. The authors should clarify that TEN2 is tagged with EGFP and that's why EGFP was measured. The authors should also better justify this experiment to provide evidence that differential extraction with saponin indeed indicate differential association with the cytoskeleton.

We have removed this experiment due to its lack of quantifiability and reproducibility.